# L-C2ST: Local Diagnostics for Posterior Approximations in Simulation-Based Inference

**Julia Linhart**
Université Paris-Saclay, Inria, CEA
Palaiseau 91120, France
julia.linhart@inria.fr

**Alexandre Gramfort**[*]
Université Paris-Saclay, Inria, CEA
Palaiseau 91120, France
alexandre.gramfort@inria.fr

**Pedro L. C. Rodrigues**
Univ. Grenoble Alpes, Inria, CNRS, Grenoble INP, LJK
Grenoble 38000, France
pedro.rodrigues@inria.fr

## Abstract

Many recent works in simulation-based inference (SBI) rely on deep generative models to approximate complex, high-dimensional posterior distributions. However, evaluating whether or not these approximations can be trusted remains a challenge. Most approaches evaluate the posterior estimator only in expectation over the observation space. This limits their interpretability and is not sufficient to identify for which observations the approximation can be trusted or should be improved. Building upon the well-known classifier two-sample test (C2ST), we introduce $\ell$-C2ST, a new method that allows for a *local* evaluation of the posterior estimator at any given observation. It offers theoretically grounded and easy to interpret – e.g. graphical – diagnostics, and unlike C2ST, does not require access to samples from the true posterior. In the case of normalizing flow-based posterior estimators, $\ell$-C2ST can be specialized to offer better statistical power, while being computationally more efficient. On standard SBI benchmarks, $\ell$-C2ST provides comparable results to C2ST and outperforms alternative local approaches such as coverage tests based on highest predictive density (HPD). We further highlight the importance of *local* evaluation and the benefit of interpretability of $\ell$-C2ST on a challenging application from computational neuroscience.

## 1 Introduction

Expressive simulators are at the core of modern experimental science, enabling the exploration of rare or challenging-to-measure events in complex systems across various fields such as population genetics [43], astrophysics [7], cosmology [32], and neuroscience [28, 16, 1, 20]. These simulators implicitly encode the intractable likelihood function $p(x \mid \theta)$ of the underlying mechanistic model, where $\theta$ represents a set of relevant parameters and $x \sim \mathrm{Simulator}(\theta)$ is the corresponding realistic observation. The main objective is to infer the parameters associated with a given observation using the simulator's posterior distribution $p(\theta \mid x)$ [4]. However, classical methods for sampling posterior distributions, such as MCMC [41] and variational inference [40], rely on the explicit evaluation of the model-likelihood, which is not possible when working with most modern simulators.

Simulation-based inference (SBI) [4] addresses this problem by estimating the posterior distribution on simulated data from the joint distribution. This can be done after choosing a prior distribution

---

[*]A. Gramfort joined Meta and can be reached at agramfort@meta.com

37th Conference on Neural Information Processing Systems (NeurIPS 2023).

$p(\theta)$ over the parameter space and using the identity $p(\theta, x) = p(x \mid \theta)p(\theta)$. In light of recent developments in the literature on deep generative models, different families of algorithms have been proposed to approximate posterior distributions in SBI [4]. Certain works use normalizing flows [37] to directly learn the posterior density function (neural posterior estimation, NPE [18]) or aim for the likelihood (neural likelihood estimation, NLE [36]). Other approaches reframe the problem in terms of a classification task and aim for likelihood ratios (neural ratio estimation, NRE [22]). However, appropriate validation remains a challenge for all these paradigms, and principled statistical approaches are still needed before SBI can become a trustworthy technology for experimental science.

This topic has been the goal of many recent studies. For instance, certain proposals aim at improving the posterior estimation by preventing over-confidence [23] or addressing model misspecification [14] to ensure conservative [8] and more robust posterior estimators [46, 27]. Another approach is the development of a SBI benchmark [31] for comparing and validating different algorithms on many standard tasks. While various validation metrics exist, Lueckmann et al. [31] show that, overall, classifier two sample tests (C2ST) [30] are currently the most powerful and flexible approach. Based on standard methods for binary classification, they can scale to high-dimensions as well as handle non-Euclidean data spaces [26, 34]. Typical use-cases include tests for statistical independence and the evaluation of sample quality for generative models [30]. Implicitly, C2ST is used in algorithms such as noise contrastive estimation [19] and generative adversarial networks [17], or to estimate likelihood-to-evidence ratios [22]. To be applied in SBI settings, however, C2ST requires access to samples from the true target posterior distribution, which renders it useless in practice. Simulation-based calibration (SBC) [42] bypasses this issue by only requiring samples from the joint distribution. Implemented in standard packages of the field (sbi [44], Stan [2]), it has become the go-to validation method for SBI [31, 23] and has been further studied in recent works [33, 5, 15]. Coverage tests based on the highest predictive density (HPD) as used in [23, 8], can be seen as a variant of SBC that are particularly well adapted to multivariate data distribution.

Nevertheless, a big limitation of current SBI diagnostics remains: they only evaluate the quality of the posterior approximation globally (in expectation) over the observation space and fail to give any insight of its *local* behavior. This hinders interpretability and can lead to false conclusions on the validity of the estimator [48, 29]. There have been attempts to make existing methods local, such as *local*-HPD [48] or *local*-multi-PIT [29], but they depend on many hyper-parameters and are computationally too expensive to be used in practice. In this work, we present $\ell$-C2ST, a new *local* validation procedure based on C2ST that can be used to evaluate the quality of SBI posterior approximations for any given observation, without using any data from the target posterior distribution. $\ell$-C2ST comes with necessary, and sufficient, conditions for the local validity of multivariate posteriors and is particularly computationally efficient when applied to validate NPE with normalizing flows, as often done in SBI literature [7, 16, 27, 46, 6, 45, 24]. Furthermore, $\ell$-C2ST offers graphical tools for analysing the inconsistency of posterior approximations, showing in which regions of the observation space the estimator should be improved and how to act upon, e.g. signs of positive / negative bias, signs of over / under dispersion, etc.

In what follows, we first introduce the SBI framework and review the basics of C2ST. Then, we detail the $\ell$-C2ST method and prove asymptotic theoretical guarantees. Finally, we report empirical results on two SBI benchmark examples to analyze the performance of $\ell$-C2ST and a non-trivial neuroscience use-case that showcases the need of a local validation method.

## 2   Validating posterior approximations with classifiers

Consider a model with parameters $\theta \in \mathbb{R}^m$ and observations $x \in \mathbb{R}^d$ obtained via a simulator. In what follows, we will always assume the typical *simulation-based inference setting*, meaning that the likelihood function $p(x \mid \theta)$ of the model cannot be easily evaluated. Given a prior distribution $p(\theta)$, it is possible to generate samples from the joint pdf $p(\theta, x)$ as per:

$$\Theta_n \sim p(\theta) \quad \Rightarrow \quad X_n = \text{Simulator}(\Theta_n) \sim p(x \mid \Theta_n) \quad \Rightarrow \quad (\Theta_n, X_n) \sim p(\theta, x) \,. \quad (1)$$

Let $N_s$ be a fixed simulation budget and $\{(\Theta_n, X_n)\}_{n=1}^{N_s} = \mathcal{D}_{\text{train}} \cup \mathcal{D}_{\text{cal}}$ with $\mathcal{D}_{\text{train}} \cap \mathcal{D}_{\text{cal}} = \emptyset$. The data from $\mathcal{D}_{\text{train}}$ are used to train an amortized[2] approximation $q(\theta \mid x) \approx p(\theta \mid x)$, e.g. via NPE [18], and those from $\mathcal{D}_{\text{cal}}$ to diagnose its *local consistency* [48].

---

[2]i.e. the approximation $q(\theta \mid x)$ is close to $p(\theta \mid x)$ on average for *all* values of $x \in \mathbb{R}^d$, so we can quickly generate samples from the posterior for any choice of conditioning observation without redoing any training.

**Definition 1** (Local consistency). *A conditional density estimator q is said to be* locally consistent at $x_o$ *with the true posterior density p if, and only if, the following null hypothesis holds:*

$$\mathcal{H}_0(x_o) : q(\theta \mid x_o) = p(\theta \mid x_o), \quad \forall \theta \in \mathbb{R}^m . \tag{2}$$

We can reformulate $\mathcal{H}_0(x_o)$ as a binary classification problem by partitioning the parameter space into two balanced classes: one for samples from the approximation ($C = 0$) and one for samples from the true posterior ($C = 1$), as in

$$\Theta \mid (C = 0) \sim q(\theta \mid x_o) \quad \text{vs.} \quad \Theta \mid (C = 1) \sim p(\theta \mid x_o) , \tag{3}$$

for which the *optimal Bayes classifier* [21] is $f^\star_{x_o}(\theta) = \operatorname{argmax}\left\{ 1 - d^\star_{x_o}(\theta), d^\star_{x_o}(\theta) \right\}$ with

$$d^\star_{x_o}(\theta) = \mathbb{P}(C = 1 \mid \Theta = \theta; x_o) = 1 - \mathbb{P}(C = 0 \mid \Theta = \theta; x_o) = \frac{p(\theta|x_o)}{p(\theta|x_o)+q(\theta|x_o)} . \tag{4}$$

It is a standard result [30, 26] to relate (2) with (3) as per

$$\mathcal{H}_0(x_o) \text{ holds} \iff d^\star_{x_o}(\theta) = \mathbb{P}(C = 1 \mid \Theta = \theta; x_o) = \tfrac{1}{2} \quad \forall \theta \tag{5}$$

When the classes are non-separable, the optimal Bayes classifier will be unable to make a decision and we can assume that it behaves as a Bernoulli random variable [30].

## 2.1 Classifier Two-Sample Test (C2ST)

The original version of C2ST [30] uses (5) to define a test statistic for $\mathcal{H}_0(x_o)$ based on the accuracy of a classifier $f_{x_o}$ trained on a dataset defined as

$$\underbrace{\Theta^q_n \sim q(\theta \mid x_o)}_{C=0} \quad \text{and} \quad \underbrace{\Theta^p_n \sim p(\theta \mid x_o)}_{C=1} \quad \text{and} \quad \mathcal{D} = \{(\Theta^q_n, 0)\}^N_{n=1} \cup \{(\Theta^p_n, 1)\}^N_{n=1} . \tag{6}$$

The classifier accuracy is then empirically estimated over $2N_v$ samples ($N_v$ samples in each class) from a held-out validation dataset $\mathcal{D}_v$ generated in the same way as $\mathcal{D}$:

$$\hat{t}_{\mathrm{Acc}}(f_{x_o}) = \frac{1}{2N_v} \sum_{n=1}^{2N_v} \left[ \mathbb{I}\left( f_{x_o}(\Theta^q_n) = 0 \right) + \mathbb{I}\left( f_{x_o}(\Theta^p_n) = 1 \right) \right]. \tag{7}$$

**Theorem 1** (Local consistency and classification accuracy). *If $f_{x_o}$ is* Bayes optimal[3] *and $N_v \to \infty$, then $\hat{t}_{\mathrm{Acc}}(f_{x_o}) = 1/2$ is a necessary and sufficient condition for the local consistency of q at $x_o$.*

See Appendix A.1 for a proof. The intuition is that, under the null hypothesis $\mathcal{H}_0(x_o)$, it is impossible for the optimal classifier to distinguish between the two data classes, and its accuracy will remain at chance-level [30]. In the context of SBI, C2ST has been used to benchmark a variety of different procedures on toy examples where the true posterior is known and can be sampled [31]. This is why we call this procedure an *oracle* C2ST, since it uses information that is not available in practice.

**Regression C2ST.** Kim et al. [26] argues that the usual C2ST based on the classifier's accuracy may lack statistical power because of the "binarization" of the posterior class probabilities. They propose to instead use probabilistic classifiers (e.g. logistic regression) of the form

$$f_{x_o}(\theta) = \mathbb{I}\left( d_{x_o}(\theta) > \tfrac{1}{2} \right) \quad \text{where} \quad d_{x_o}(\theta) = \mathbb{P}(C = 1 \mid \theta; x_o) \tag{8}$$

and define the test statistics in terms of the predicted class probabilities $d_{x_o}$ instead of the predicted class labels. The test statistic is then the mean squared distance between the estimated class posterior probability and one half:

$$\hat{t}_{\mathrm{MSE}}(f_{x_o}) = \frac{1}{N_v} \sum_{n=1}^{N_v} \left( d_{x_o}(\Theta^q_n) - \frac{1}{2} \right)^2 + \frac{1}{N_v} \sum_{n=1}^{N_v} \left( d_{x_o}(\Theta^p_n) - \frac{1}{2} \right)^2 \tag{9}$$

**Theorem 2** (Local consistency and regression). *If $f_{x_o}$ is* Bayes optimal *and $N_v \to \infty$, then $\hat{t}_{\mathrm{MSE}}(f_{x_o}) = 0$ is a necessary and sufficient condition for the local consistency of q at $x_o$.*

See Appendix A.2 for a proof. The numerical illustrations in Kim et al. [26] give empirical evidence that *Regression C2ST* has superior statistical power as compared to its accuracy-based counterpart, particularly for high-dimensional data spaces. Furthermore, it offers tools for interpretation and visualization: evaluating the predicted class probabilities $d_{x_o}(\theta)$ for any $\theta \in \mathbb{R}^m$ informs the regions where the classifier is more (or less) confident about its choice [30, 26].

---

[3]i.e. it is the classifier with lowest possible classification error for the dataset $\mathcal{D}$.

# 3   $\ell$-C2ST: Local Classifier Two-Sample Tests

The *oracle* C2ST framework is not applicable in practical SBI settings, since it requires access to samples from the true posterior distribution to (1) **train** a classifier and (2) **evaluate** its performance in discriminating data from $q$ and $p$. This section presents a new method called *local* C2ST ($\ell$-C2ST) capable of evaluating the local consistency of a posterior approximation requiring data only from the joint pdf $p(\theta, x)$ which can be easily sampled as per (1).

**(1) Train the classifier.** We define a modified version of the classification framework (3) with:

$$(\Theta, X) \mid (C = 0) \sim q(\theta \mid x)p(x) \quad \text{vs.} \quad (\Theta, X) \mid (C = 1) \sim p(\theta, x) . \tag{10}$$

The optimal Bayes classifier is now $f^\star(\theta, x) = \operatorname{argmax}\{1 - d^\star(\theta, x), d^\star(\theta, x)\}$ with

$$d^\star(\theta, x) = \frac{p(\theta, x)}{p(\theta, x) + q(\theta \mid x)p(x)} = \frac{p(\theta \mid x)}{p(\theta \mid x) + q(\theta \mid x)} = d^\star_x(\theta) , \tag{11}$$

where one can notice the direct relation with the Bayes classifier for (3). Therefore, using data sampled as in (10), it is possible to train a classifier $f(\theta, x)$ and write $f_{x_\mathrm{o}}(\theta) = f(\theta, x_\mathrm{o})$ for each $x_\mathrm{o}$. See Algorithm 1 for details on the implementation of this procedure.

**(2) Evaluate the classifier.** Define a new test statistic that evaluates the MSE-statistic for a classifier $f$ and its associated predicted probabilities $d$ using data samples from only the class associated to the posterior approximation ($C = 0$):

$$\hat{t}_{\mathrm{MSE}_0}(f, x_\mathrm{o}) = \frac{1}{N_v} \sum_{n=1}^{N_v} \left( d(\Theta_n^q, x_\mathrm{o}) - \frac{1}{2} \right)^2 \quad \text{with} \quad \Theta_n^q \sim q(\theta \mid x_\mathrm{o}) . \tag{12}$$

**Theorem 3** (Local consistency and single class evaluation). *If $f$ is Bayes optimal and $N_v \to \infty$, then $\hat{t}_{\mathrm{MSE}_0}(f, x_\mathrm{o}) = 0$ is a necessary and sufficient condition for the local consistency of $q$ at $x_\mathrm{o}$.*

*Proof.* Let $d$ be an estimator of $\mathbb{P}(C = 1 \mid \Theta, X)$ and $f = \mathbb{I}(d > 0.5)$. Suppose that $f = f^\star$ is *Bayes optimal* and let $x_\mathrm{o}$ be a fixed observation. We have that

$$\lim_{N_v \to \infty} \hat{t}_{\mathrm{MSE}_0}(f^\star, x_\mathrm{o}) = \int \left( d^\star_{x_\mathrm{o}}(\theta) - \frac{1}{2} \right)^2 q(\theta \mid x_\mathrm{o}) \mathrm{d}\theta .$$

Because of the squared term in the integral and $q$ being a p.d.f., we have that

$$\lim_{N_v \to \infty} \hat{t}_{\mathrm{MSE}_0}(f^\star, x_\mathrm{o}) = 0 \quad \Longleftrightarrow \quad d^\star_{x_\mathrm{o}}(\theta) = \mathbb{P}(C = 1 \mid \theta; x_\mathrm{o}) = \tfrac{1}{2} .$$

This new statistical test can thus be used to assess the local consistency of posterior approximation $q$ without using any sample from the true posterior distribution $p$, but only from the joint pdf. Furthermore, it is amortized, so a single classifier is trained for (10) that can then be used for any choice of conditioning observation $x_\mathrm{o}$. This is not the case in the usual *oracle* C2ST framework.

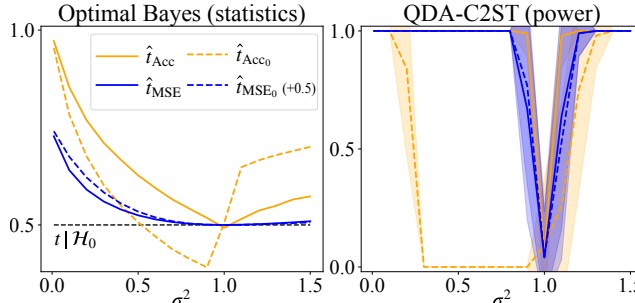

Figure 1: Results for the C2ST framework when $p = \mathcal{N}(0, \mathbf{I}_2)$ and $q = \mathcal{N}(0, \sigma^2 \mathbf{I}_2)$. **Left** panel portrays the test statistics for the optimal Bayes classifier and the **right** panel shows the test's empirical power with QDA. Single-class accuracy test ($\hat{t}_{\mathrm{Acc}_0}$) fails to detect when $p \neq q$ but $\hat{t}_{\mathrm{MSE}_0}$ behaves correctly.

Figure 1 illustrates the behavior of different test statistics to discriminate samples from two bivariate Normal distributions whose covariance mismatch is controlled by a single scaling parameter $\sigma$. Note that the optimal Bayes classifier for this setting can be obtained via quadratic discriminant analysis (QDA) [21]. The results clearly show that even though $t_{\mathrm{MSE}_0}$ exploits only half of the dataset (i.e. samples from class $C = 0$) it is capable of detecting when $p$ and $q$ are different ($\sigma \neq 1$). The plot also

includes the results for a one-class test statistic based on accuracy values ($\hat{t}_{\text{Acc}_0}$) which, as opposed to $\hat{t}_{\text{MSE}_0}$, has no guarantees for being a necessary and sufficient condition for local consistency. Not surprisingly, it fails to reject the null hypothesis for various choices of $\sigma$.

The assumptions of Theorem 3 are never met in practice: datasets are finite and one rarely knows which type of classifier is optimal for a given problem. Therefore, the values of $\hat{t}_{\text{MSE}_0}$ in the null hypothesis ($p = q$) tend to fluctuate around one-half, and it is essential to determine a threshold for deciding whether or not $\mathcal{H}_0(x_o)$ should be rejected. In $\ell$-C2ST, these threshold values are obtained via a permutation procedure [13] described in Algorithm 1. This yields $N_{\mathcal{H}}$ estimates of the test statistic under the null hypothesis and can be used to calculate $p$-values for any given $\alpha$ significance level as described in Algorithm 2. These estimates can also be used to form graphical summaries known as PP-plots, which display the empirical CDF of the probability predictions versus the nominal probability level. These plots show how the predicted class probability $d(x_o)$ deviates from its theoretical value under the null hypothesis (i.e. one half) as well as $(1 - \alpha)$ confidence regions; see Algorithm B.1 available in the appendix for more details and Figure 4 for an example.

---

**Algorithm 1:** $\ell$-C2ST – training the classifier on data from the joint distribution

---

**Input:** posterior estimator $q$; calibration data $\mathcal{D}_{\text{cal}} = \{\Theta_n, X_n\}_{n=1}^{N_{\text{cal}}}$; classifier $f$; number of samples $N_{\mathcal{H}}$ from the distribution under the null hypothesis
**Output:** estimate $d$ of the class probabilities; estimates $\{d_1, \ldots, d_{N_{\mathcal{H}}}\}$ under the null hypothesis
/* Construct classification training set                                          */
**for** $n = 1, \ldots, N_{\text{cal}}$ **do**
$\quad$ $\Theta_n^q \sim q(\theta \mid X_n)$
$\quad$ $W_{2n} = (\Theta_n^q, X_n); C_{2n} = 0$ /* Sample from $q(\theta \mid x)p(x)$       */
$\quad$ $W_{2n+1} = (\Theta_n, X_n); C_{2n+1} = 1$ /* Sample from $p(\theta, x)$       */
$\mathcal{D} \leftarrow \{W_n, C_n\}_{n=1}^{2N_{\text{cal}}}$
/* Get estimate $d$ of the class probabilities                                    */
Train the classifier $f$ on $\mathcal{D}$
$d \leftarrow f_{\text{probability}}$
/* Estimate $d$ under the null hypothesis via permutation procedure       */
**for** $h = 1, \ldots N_{\mathcal{H}}$ **do**
$\quad$ Randomly permute labels $C_n$ in $\mathcal{D}$
$\quad$ Train the classifier $f$ on new $\mathcal{D}$
$\quad$ $d_h \leftarrow f_{\text{probability}}$
**return** $d; \{d_1, \ldots, d_{N_{\mathcal{H}}}\}$

---

**Algorithm 2:** $\ell$-C2ST – evaluating test statistics and $p$-values for any $x_o$

---

**Input:** Observation $x_o$; estimates $d$ and $\{d_1, \ldots, d_{N_{\mathcal{H}}}\}$ obtained in Algorithm 1
**Output:** test statistic $\hat{t}_{\text{MSE}_0}(x_o)$; p-value $\hat{p}(x_o)$
Generate $N_v$ samples $\Theta_n^q \sim q(\theta \mid x_o)$ with predicted probabilities $d(\Theta_n^q, x_o)$ and $d_h(\Theta_n^q, x_o)$
/* Compute test statistics                                                        */
$\hat{t}_{\text{MSE}_0}(x_o) \leftarrow \frac{1}{N_v} \sum_n \left(d(\Theta_n^q, x_o) - \frac{1}{2}\right)^2$
**for** $h = 1, \ldots, N_{\mathcal{H}}$ **do**
$\quad$ $\hat{t}_h(x_o) \leftarrow \frac{1}{N_v} \sum_n \left(d_h(\Theta_n^q, x_o) - \frac{1}{2}\right)^2$
/* Compute p-value                                                                */
$\hat{p}(x_o) \leftarrow \frac{1}{N_{\mathcal{H}}} \sum_h \mathbb{I}\left(\hat{t}_h(x_o) > \hat{t}_{\text{MSE}_0}(x_o)\right)$
**return** $\hat{t}_{\text{MSE}_0}(x_o), \hat{p}(x_o)$

---

### 3.1 The case of normalizing flows

The $\ell$-C2ST framework can be further improved when the posterior approximation $q$ is a conditional normalizing flow [37], which we denote $q_\phi$. Given a Gaussian base distribution $u(z) = \mathcal{N}(0, I_m)$ and a bijective transform $T_\phi(\cdot; x)$ with Jacobian $J_{T_\phi}(\cdot; x)$ we have

$$q_\phi(\theta \mid x) = u(z)|\det J_{T_\phi}(z; x)|^{-1}, \quad \theta = T_\phi(z; x) \in \mathbb{R}^m \ . \tag{13}$$

In other words, normalizing flows (NF) are invertible neural networks that define a map between a latent space where data follows a Gaussian distribution and the parameter space containing complex posterior distributions. This allows for both efficient sampling and density evaluation:

$$Z \sim \mathcal{N}(0, I_m) \;\Rightarrow\; \Theta^q = T_\phi(Z; x) \sim q_\phi(\theta \mid x) \,, \tag{14}$$

$$q_\phi(\theta \mid x) = u(T_\phi^{-1}(\theta; x)) |\det J_{T_\phi^{-1}}(\theta; x)| \,. \tag{15}$$

Our main observation is that the inverse transform $T_\phi^{-1}$ can also be used to characterize the local consistency of the conditional normalizing flow in its latent space, yielding a much simpler and computationally less expensive statistical test for posterior local consistency.

**Theorem 4** (Local consistency and normalizing flows). *Given a posterior approximation $q_\phi$ based on a normalizing flow, its local consistency at $x_\mathrm{o}$ can be characterized as follows:*

$$p(\theta \mid x_\mathrm{o}) = q_\phi(\theta \mid x_\mathrm{o}) \iff p(T_\phi^{-1}(\theta; x_\mathrm{o}) \mid x_\mathrm{o}) = u(z) \,, \quad \forall \theta \in \mathbb{R}^m \,. \tag{16}$$

*Proof.* Let $\Theta \sim p(\theta \mid x_\mathrm{o})$. Following (14), we have that $\Theta \sim q_\phi(\theta \mid x_\mathrm{o})$ if, and only if, $\Theta = T_\phi(Z; x_\mathrm{o})$ with $Z \sim \mathcal{N}(0, I_m)$. Applying the inverse transformation of the flow gives us $T_\phi^{-1}(\Theta; x_\mathrm{o}) = T_\phi^{-1}(T_\phi(Z; x_\mathrm{o}); x_\mathrm{o}) = Z \sim \mathcal{N}(0, I_m)$, which concludes the proof.

Based on Theorem 4 we propose a modified version of our statistical test named $\ell$-C2ST-NF. The new null hypothesis associated with the consistency of the posterior approximation $q_\phi$ at $x_\mathrm{o}$ is

$$\mathcal{H}_0^{\mathrm{NF}}(x_\mathrm{o}) : p(T_\phi^{-1}(\theta; x_\mathrm{o}) \mid x_\mathrm{o}) = \mathcal{N}(0, I_m) \,, \tag{17}$$

which leads to a new binary classification framework

$$(Z, X) \mid (C = 0) \sim \mathcal{N}(0, I_m)p(x) \quad \text{vs.} \quad (Z, X) \mid (C = 1) \sim p(T_\phi^{-1}(\theta; x), x) \,. \tag{18}$$

Algorithm 3 describes how to sample data from each class and **train** a classifier to discriminate them. The classifier is then **evaluated** on $N_v$ samples $Z_n \sim \mathcal{N}(0, I_m)$ which are independent of $x_\mathrm{o}$.

A remarkable feature of $\ell$-C2ST-NF is that calculating the test statistics under the null hypothesis is considerably faster than for $\ell$-C2ST. In fact, for each null trial $h = 1, \ldots, N_{\mathcal{H}}$ we use the dataset $\mathcal{D}_{\mathrm{cal}}$ only for recovering the samples $X_n$ and then *independently* sample new data $Z_n \sim \mathcal{N}(0, I_m)$. As such, it is possible to pre-train the classifiers without relying on a permutation procedure (cf. Algorithm 4), and to re-use them to quickly compute the validation diagnostics for any choice of $x_\mathrm{o}$ or new posterior estimator $q_\phi$ of the given inference task. It is worth mentioning that this is not possible with the usual $\ell$-C2ST, as it depends on $q$ and it would require new simulations for each trial.

---

**Algorithm 3:** $\ell$-C2ST-NF – training the classifier on the joint distribution

---

**Input:** NF posterior estimator $q_\phi$; calibration data $\mathcal{D}_{\mathrm{cal}} = \{\Theta_n, X_n\}_{n=1}^{N_{\mathrm{cal}}}$ ; classifier $f$
**Output:** estimate $d$ of the class probabilities
/* Construct classification training set                                    */
**for** $n$ *in* $1, \ldots, N_{\mathrm{cal}}$ **do**
    $Z_n \sim \mathcal{N}(0, I_m);\ Z_n^q = T_\phi^{-1}(\Theta_n; X_n)$ /* inverse NF-transformation              */
    $W_{2n} = (Z_n, X_n);\ C_{2n} = 0$
    $W_{2n+1} = (Z_n^q, X_n);\ C_{2n+1} = 1$
$\mathcal{D} \leftarrow \{W_n, C_n\}_{n=1}^{2N_{\mathrm{cal}}}$
/* Get estimate $d$ of the class probabilities                              */
Train the classifier $f$ on $\mathcal{D}$
$d \leftarrow f_{\mathrm{probability}}$
**return** $d$

---

**Algorithm 4:** $\ell$-C2ST-NF – precompute the null distribution for any estimator

---

**Input:** calibration data $\mathcal{D}_{\text{cal}} = \{\Theta_n, X_n\}_{n=1}^{N_{\text{cal}}}$; classifier $f$; number of null samples $N_{\mathcal{H}}$
**Output:** estimates $\{d_1, \ldots, d_{N_{\mathcal{H}}}\}$ of the class probabilities under the null
**for** $h$ *in* $1, \ldots N_{\mathcal{H}}$ **do**

> /* Construct classification training set                         */
> Sample $Z_n \sim \mathcal{N}(0, I_m)$ for $n = 1, \ldots, 2N_{\text{cal}}$
> $\mathcal{D} \leftarrow \{(Z_{2n}, X_n), 0\}_{n=1}^{N_{\text{cal}}} \cup \{(Z_{2n+1}, X_n), 1\}_{n=1}^{N_{\text{cal}}}$
> /* Get estimate $d$ of the class probabilities             */
> Train the classifier $f$ on $\mathcal{D}$
> $d_h \leftarrow f_{\text{probability}}$

**return** $\{d_1, \ldots, d_{N_{\mathcal{H}}}\}$

---

## 4 Experiments

All experiments were implemented with Python and the `sbi` package [44] combined with PyTorch [38] and `nflows` [12] for neural posterior estimation [4]. Classifiers on the C2ST framework use the `MLPClassifier` from scikit-learn [39] with the same parameters as those used in `sbibm` [31].

### 4.1 Two benchmark examples for SBI

We illustrate $\ell$-C2ST on two examples: `Two Moons` and SLCP. These models have been widely used in previous works from SBI literature [18, 36] and are part of the SBI benchmark [31]. They both represent difficult inference tasks with locally structured multimodal true posteriors in, respectively, low ($\theta \in \mathbb{R}^2, x \in \mathbb{R}^2$) and high ($\theta \in \mathbb{R}^5, x \in \mathbb{R}^8$) dimensions. See [31] for more details. To demonstrate the benefits of $\ell$-C2ST-NF, all experiments use neural spline flows [11] trained under the amortized paradigm for neural posterior estimation (NPE) [37]. We use implementations from the `sbibm` package [31] to ensure uniform and consistent experimental setups. Samples from the true posterior distributions for both examples are obtained via MCMC and used to compare $\ell$-C2ST(NF) to the *oracle*-C2ST framework. We include results for *local*-HPD implemented using the code repository of the authors of [48] with default hyper-parameters and applied to HPD.[5]

First, we evaluate the local consistency of the posterior estimators of each task over ten different observations $x_{\text{o}}$ with varying $N_{\text{train}}$ and fixed $N_{\text{cal}} = 10^4$. The first column of Figure 2 displays the MSE statistics for the *oracle* and $\ell$-C2ST frameworks. As expected, we observe a decrease in the test statistics as $N_{\text{train}}$ increases: more training data usually means better approximations. For `Two Moons` the statistic of $\ell$-C2ST decreases at the same rate as *oracle*-C2ST, with notably accurate results for $\ell$-C2ST-NF. However, in SLCP both $\ell$-C2ST statistics decrease much faster than the *oracle*. This is possibly due to the higher dimension of the observation space in the latter case, which impacts the training-procedure of $\ell$-C2ST on the joint distribution.

We proceed with an empirical analysis based on 50 random test runs for each validation method and computing their rejection-rates to the nominal significance level of $\alpha = 0.05$. Column 4 in Figure 2 confirms that the false positive rates (or type I errors) for all tests are controlled at desired level. Column 2 of Figure 2 portrays the true positive rates (TPR), i.e. rejecting $\mathcal{H}_0(x_{\text{o}})$ when $p \neq q$, of each test as a function of $N_{\text{train}}$. Both $\ell$-C2ST strategies decrease with $N_{\text{train}}$ as in Column 1, with higher TPR for $\ell$-C2ST-NF in both tasks. The *oracle* has maximum TPR and rejects the local consistency of all posterior estimates across all observations at least 90% of the time. Note that SLCP is designed to be a difficult model to estimate[6], meaning that higher values of TPR are expected ($\hat{t} \not\to 0$ in Column 1). In `Two Moons`, the decreasing rejection rate can be seen as normal as it reflects the convergence of the posterior approximator ($\hat{t} \to 0$ in Column 1): as $N_{\text{train}}$ increases, the task of differentiating the estimator from the true posterior becomes increasingly difficult.

We fix $N_{\text{train}} = 10^3$ (which yields inconsistent $q_\phi$ in both examples) and investigate in Column 3 of Figure 2 how many calibration samples are needed to get a maximal TPR in each validation method.

---

[4]Code is available at https://github.com/JuliaLinhart/lc2st.
[5]The average run-times for each validation method are provided in Appendix C.
[6]SLCP = simple likelihood, complex posterior

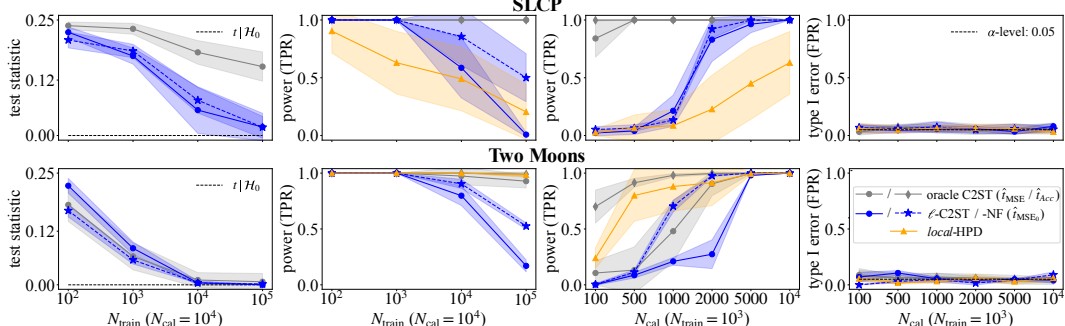

Figure 2: Results on two examples from the SBI benchmark: `SLCP` and `Two Moons`. We compare $\ell$-C2ST and $\ell$-C2ST-NF (dashed) to the *oracle* C2ST and *local*-HPD. Columns 1 and 2 display the test statistic and empirical power as a function of $N_{\text{train}}$, while Columns 3 and 4 show the empirical power and type I error for varying $N_{\text{cal}}$. The $\ell$-C2ST(-NF) statistics are comparable to the oracle, as their decreasing behaviour reflects the convergence of NPE to the true posterior for large training datasets. We also note that $\ell$-C2ST-NF is uniformly better than $\ell$-C2ST (i.e. higher power for all $N_{\text{train}}$ and increases faster with $N_{\text{cal}}$), and both reach maximum statistical power with smaller $N_{\text{cal}}$ than *local*-HPD. All Type I errors are controlled at $\alpha = 0.05$. Experiments were performed over 10 different observations $x_{\text{o}}$ (mean and std) and Columns 2-4 used additional 50 random test runs.

`SLCP` is expected to be an easy classification task, since the posterior estimator is very far from the true posterior (large values of $\hat{t}$ in Column 1). We observe similar performance for $\ell$-C2ST-NF and $\ell$-C2ST, with slightly faster convergence for the latter. Both methods perform better than *local*-HPD, that never reaches maximum TPR. `Two Moons` represents a harder discrimination task, as $q_\phi$ is already pretty close to the reference posterior (see Column 1). Here, $\ell$-C2ST-NF attains maximum power at $N_{\text{cal}} = 2000$ and outperforms all other methods. Surprisingly, the regression-based *oracle*-C2ST performs comparably to *local*-HPD, converging to TPR = 1 at $N_{\text{cal}} = 5000$.

## 4.2 Jansen-Rit Neural Mass Model (JRNMM)

We increase the complexity of our examples and consider the well known Jansen & Rit neural mass model (JRNMM) [25]. This is a neuroscience model which takes parameters $\boldsymbol{\theta} = (C, \mu, \sigma, g) \in \mathbb{R}^4$ as input and generates time series $x \in \mathbb{R}^{1024}$ with properties similar to brain signals obtained in neurophysiology. Each parameter has a physiologically meaningful interpretation, but they are not relevant for the purposes of this section; the interested reader is referred to [3] for more details.

The approximation $q_\phi$ of the model's posterior distribution is a conditioned masked autoregressive flow (MAF) [35] with 10 layers. We follow the same experimental setting from [3], with a uniform prior over physiologically-relevant parameter values and a simulated dataset from the joint distribution including $N_{\text{train}} = 50\,000$ training samples for the posterior estimator and $N_{\text{cal}} = 10\,000$ samples to compute the validation diagnostics. An evaluation set of size $N_{v0} = 10\,000$ is used for $\ell$-C2ST-NF.

We first investigate the *global consistency* of our approximation, which informs whether $q_\phi$ is consistent (or not) on average with the model's true posterior distribution. We use standard tools for this task such as simulation-based calibration (SBC) [42] and coverage tests based on highest predictive density (HPD) [48]. The results are shown in the left panel of Figure 3. We observe that the empirical cdf of the global HPD rank-statistic deviates from the identity function (black dashed line), indicating that the approximator presents signs of global inconsistency. We also note that the marginal SBC-ranks are unable to detect any inconsistencies in $q_\phi$.

We use $\ell$-C2ST-NF to study the *local consistency* of $q_\phi$ on a set of nine observations $x_{\text{o}}^{(i)}$ defined as[7]

$$x_{\text{o}}^{(i)} = \text{JRNMM}(\theta_{\text{o}}^{(i)}) \quad \text{with} \quad \theta_{\text{o}}^{(i)} = (135, 220, 2000, g_{\text{o}}^{(i)}) \quad \text{and} \quad g_{\text{o}}^{(i)} \in [-20, +20] \,. \quad (19)$$

The right panel of Figure 3 shows that the test statistics of $\ell$-C2ST-NF vary in a U-shape, attaining higher values as $g_{\text{o}}$ deviates from zero and at the borders of the prior. The plot is overlaid with the 95% confidence region, illustrating how much the test statistics deviate from the local null hypothesis.

---

[7]Note that the uniform prior for $g$ is defined in [-20, +20] when training $q_\phi$ with NPE.

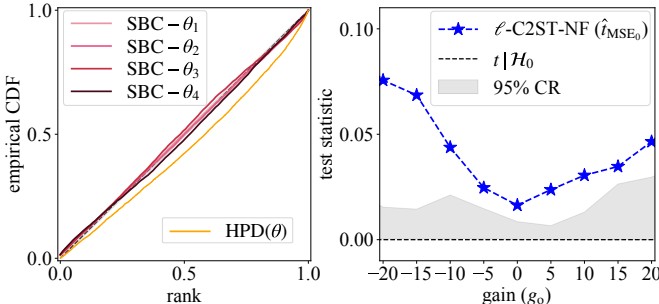

Figure 3: Results for global and local tests on JRNMM. **Left**: PP-plots for the marginal SBC and global HPD rank statistics. **Right**: Test statistics for $\ell$-C2ST-NF on observations with varying $g_\mathrm{o}$. SBC fails to detect any inconsistency of $q_\phi$, while HPD only provides a global assessment, unlike $\ell$-C2ST which locally explains the inconsistencies in $q_\phi$.

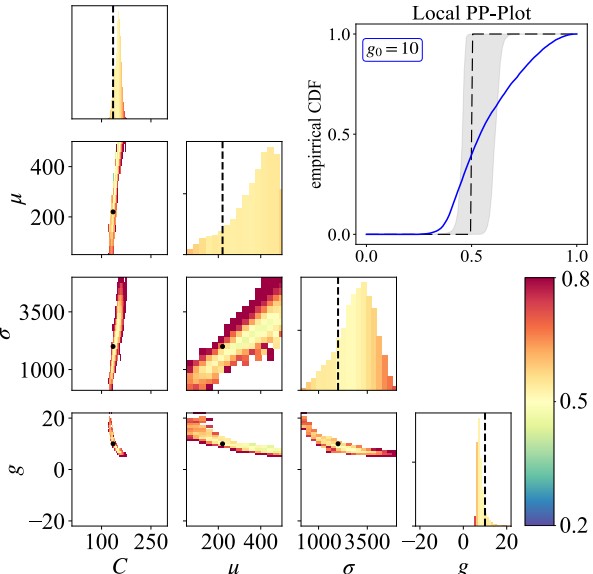

Figure 4: Graphical diagnostics of $\ell$-C2ST-NF for JRNMM. Top right panel displays the empirical CDF of the classifier (blue) overlaid with the theoretical CDF of the null hypothesis (step function at $0.5$) and $95\%$ confidence region of estimated classifiers under the null displayed in gray. The pairplot displays histograms of samples from $q_\phi$ within the prior region and dashed lines indicate values of $\theta_\mathrm{o}$ used to generate the conditioning observation $x_\mathrm{o}$. The predicted probabilities are mapped on the colors of the bins in the histogram. Blue-green (resp. orange-red) regions indicate low (resp. high) predicted probabilities of the classifier. Yellow regions correspond to chance level, thus $q_\phi \approx p$.

We demonstrate the interpretability of the results for $\ell$-C2ST-NF with a focus on the behavior of $q_\phi$ when conditioned on an observation for which $g_\mathrm{o} = 10$. The local PP-plot in Figure 4 summarises the test result: the predicted class probabilities deviate from $0.5$, outside of the $95\%$-CR, thus rejecting the null hypothesis of local consistency at $g_\mathrm{o} = 10$. The rest of Figure 4 displays 1D and 2D histograms of samples $\Theta^q \sim q_\phi(\theta \mid x_\mathrm{o})$ within the prior region, obtained by applying the learned $T_\phi$ to samples $Z \sim \mathcal{N}(0, I_4)$. The color of each histogram bin is mapped to the intensity of the corresponding predicted probability in $\ell$-C2ST-NF and informs the regions where the classifier is more (resp. less) confident about its choice of predicting class $0$.[8] This relates to regions in the parameter space where the posterior approximation has too much (resp. not enough mass) w.r.t. to the true posterior: $q_\phi > p$ (resp. $q_\phi < p$). We observe that the ground-truth parameters are often outside of the red regions, indicating positive bias for $\mu$ and $\sigma$ and negative bias for $g$ in the 1D marginal. It also shows that the posterior is over-dispersed in all 2D marginals. See Appendix D for results on all observations $x_\mathrm{o}^{(i)}$.

## 5 Discussion

We have presented $\ell$-C2ST, an extension to the C2ST framework tailored for SBI settings which requires only samples from the joint distribution and is amortized along conditioning observations. Strikingly, empirical results show that, while $\ell$-C2ST does not have access to samples from the true posterior distribution, it is actually competitive with the oracle-C2ST approach that does. This comes at the price of training a binary classifier on a potentially large dataset to ensure the correct calibration

---

[8]Specifically, we compute the *average* predicted probability of class 0 for data points $Z \sim \mathcal{N}(0, I_4)$ corresponding to samples $T_\phi(Z; x_\mathrm{o}) \sim q_\phi(\theta \mid x_\mathrm{o})$ within each histogram bin.

of the predicted probabilities. Should this be not the case, some additional calibration step for the classifier can be considered [10].

Notably, $\ell$-C2ST allows for a local analysis of the consistency of posterior approximations and is more sensible, precise, and computationally efficient than its concurrent method, *local*-HPD. Appendix F.4 provides a detailed discussion of these statements, based on results obtained for additional benchmark examples. When exploiting properties of normalizing flows, $\ell$-C2ST can be further improved as demonstrated by encouraging results on difficult posterior estimation tasks (see Table 2 in Appendix F). We further analyze the benefits of this -NF version in Appendix F.3. $\ell$-C2ST provides necessary, and sufficient, conditions for posterior consistency, features that are not shared by other standard methods in the literature (e.g. SBC). When applied to a widely used model from computational neuroscience, the local diagnostic proposed by $\ell$-C2ST offered interesting and useful insights on the failure modes of the SBI approach (e.g. poor estimates on the border of the prior intervals), hence demonstrating its potential practical relevance for works leveraging simulators for scientific discoveries.

## 6    Limitations and Perspectives

**Training a classifier with finite data.** The proposed validation method leverages classifiers to learn global and local data structures and shows great potential in diagnosing conditional density estimators. However, it's validity is only theoretically guaranteed by the optimality of the classifier when $N_v \rightarrow \infty$. In practice, this can never perfectly be ensured. Figure 6 in Appendix F.2 shows that depending on the dataset, $\ell$-C2ST can be more or less accurate w.r.t. the true C2ST. Therefore, one should always be concerned about false conclusions due to a far-from-optimal classifier and make sure that the classifier is "good enough" before using it as a diagnostic tool, e.g. via cross-validation. Note, however, that the MSE test statistic for $\ell$-C2ST is defined by the predicted class probabilities and not the accuracy of the classifier, thus one should also check how well the classifier is calibrated.

**Why Binary Classification?** An alternative to binary classification would have been to train a second posterior estimate in order to assess the consistency of the first one. Indeed, one could ask whether training a classifier is inherently easier than obtaining a good variational posterior, the response to which is non-trivial. Nevertheless, we believe that adding diversity into the validation pipeline with two different ML approaches might be preferable. Furthermore, building our method on the C2ST framework was mainly motivated by the popularity and robustness of binary classification: it is easy to understand and has a much richer and stable literature than deep generative models. As such, we believe that choosing a validation based on a binary classifier has the potential of attracting the interest of scientists across various fields, rather than solely appealing to the probabilistic machine learning community.

**Possible extensions and improvements.** Future work could focus on leveraging additional information of $q$ while training the classifier as in [47]. For example, by using more samples from the posterior estimator $q$ or its explicit likelihood function (which is accessible when $q$ is a normalizing flow). On a different note, split-sample conformal inference could be used to speed up the $p$-value calculations (avoiding the time-consuming permutation step in Algorithm 1).

In summary, our article shows that $\ell$-C2ST is theoretically valid and works on several datasets, sometimes even outperforming *local*-HPD, which to our knowlegde is the only other existing local diagnostic. Despite facing some difficulties for certain examples (just like for other methods as well), an important feature of $\ell$-C2ST is that one can directly leverage from improvements in binary classification to adapt and enhance it for any given dataset and task. This makes $\ell$-C2ST a competitive alternative to other validation approches, with great potential of becoming the go-to validation diagnostic for SBI practitioners.

## Acknowledgments

Julia Linhart is recipient of the Pierre-Aguilar Scholarship and thankful for the funding of the Capital Fund Management (CFM). Alexandre Gramfort was supported by the ANR BrAIN (ANR-20-CHIA0016) grant while in his role at Université Paris-Saclay, Inria.

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

# Appendices

# A Proofs

*In what follows, we assume that it is sufficient for the null hypothesis $\mathcal{H}_0$ to hold on any set $\mathcal{C} \subseteq \mathbb{R}^m$ of strictly positive measure, rather than requiring it to hold for all points $\theta \in \mathbb{R}^m$. In practice, it generally has no practical implications if the posterior estimator is inconsistent with the true posterior ($q \neq p$) on a set of measure zero, since those sets don't have any real statistical significance.*

*For the proofs of Theorems 1 and 2, we will consider a classifier $f_{x_o}$ defined for a fixed observation $x_o \in \mathbb{R}^d$ on $\mathcal{S}_{x_o} = \{\theta \in R^m, q(\theta \mid x_o) + p(\theta \mid x_o) > 0\}$.*

## A.1 Proof of Theorem 1

**Theorem 1** (Local consistency and classification accuracy). If $f_{x_o}$ is *Bayes optimal* and $N_v \to \infty$, then $\hat{t}_{\mathrm{Acc}}(f_{x_o}) = 1/2$ is a necessary and sufficient condition for the local consistency of $q$ at $x_o$.

*Proof.* As $\mathcal{S}_{x_o}$ contains all data points $\Theta_n^q \sim q(\theta \mid x_o)$ ($C_n = 0$) and $\Theta_n^p \sim p(\theta \mid x_o)$ ($C_n = 1$), the empirical accuracy $\hat{t}_{\mathrm{Acc}}(f_{x_o})$ over the validation set $\mathcal{D}_v = \{(\Theta_n, C_n)\}_{n=1}^{2N_v}$ is well defined (7) and

$$\hat{t}_{\mathrm{Acc}}(f_{x_o}) \xrightarrow[N_v \to \infty]{} \mathrm{Acc}(f_{x_o}) = \mathrm{P}(f_{x_o}(\Theta) = C) = \frac{1}{2}\left(\mathrm{P}(f_{x_o}(\Theta^q) = 0) + \mathrm{P}(f_{x_o}(\Theta^p) = 1)\right) .$$

Let's show that if $f_{x_o}$ is Bayes optimal, then $\mathcal{H}_0(x_o)$ holds $\iff \mathrm{Acc}(f_{x_o}) = \frac{1}{2}$.

($\Rightarrow$): Suppose $\mathcal{H}_0(x_o)$ holds. The optimality of $f_{x_o}$ implies that $f_{x_o}(\Theta^q)$ and $f_{x_o}(\Theta^p)$ are Bernoulli random variables $\mathcal{B}(\frac{1}{2})$ (see interpretation of equation (5)), and so $\mathrm{Acc}(f_{x_o}) = \frac{1}{2}\left(\frac{1}{2} + \frac{1}{2}\right) = \frac{1}{2}$.

($\Leftarrow$): Let's proceed by showing the contraposition: if $\mathcal{H}_0(x_o)$ does not hold, then $\mathrm{Acc}(f_{x_o}) \neq \frac{1}{2}$.

Suppose that $\mathcal{H}_0(x_o)$ does not hold, there exists a set $\mathcal{C} = \left\{\theta \in \mathbb{R}^m, p(\theta \mid x_o) \neq q(\theta \mid x_o)\right\}$ of strictly positive measure (w.r.t. $p$ or $q$, which ever is non zero on that set). We can decompose $\mathcal{C}$ into the direct sum of $\mathcal{A} = \left\{\theta \in \mathcal{C}, q(\theta \mid x_o) < p(\theta \mid x_o)\right\}$ and $\mathcal{B} = \left\{\theta \in \mathcal{C}, q(\theta \mid x_o) > p(\theta \mid x_o)\right\}$. Either $\mathcal{A}$ or $\mathcal{B}$ is necessarily of strictly positive measure. Let's say $\mathcal{A}$ (see Lemma 1 for the symmetric case).

$\mathcal{A}$ is exactly the region where $\mathrm{Prob}(C = 1 \mid \Theta = \theta) > \frac{1}{2}$ and thus where $f_{x_o}(\theta) = 1$; $\mathcal{B}$ is the region where $f_{x_o}(\theta) = 0$. We therefore get that:

$$\begin{aligned}
\mathrm{Acc}(f_{x_o}) &= \frac{1}{2}\left(\int_{f_{x_o}(\theta)=0} q(\theta \mid x_o)\mathrm{d}\theta + \int_{f_{x_o}(\theta)=1} p(\theta \mid x_o)\mathrm{d}\theta\right) \\
&= \frac{1}{2}\left(\int_{\mathcal{B}} q(\theta \mid x_o)\mathrm{d}\theta + \int_{\mathcal{A}} p(\theta \mid x_o)\mathrm{d}\theta\right) \\
&= \frac{1}{2}\left(1 + \int_{\mathcal{A}} p(\theta \mid x_o) - q(\theta \mid x_o)\mathrm{d}\theta\right) \quad \text{(because } \int_{\mathcal{A}} q + \int_{\mathcal{B}} q = 1\text{)}
\end{aligned}$$

But $\forall \theta \in \mathcal{A}$, $0 \leq q(\theta \mid x) < p(\theta \mid x)$ (and $\mathcal{A}$ is of strictly positive measure), so the integral in the last equality is strictly positive and we get $\mathrm{Acc}(f_x) > \frac{1}{2}$, which concludes the proof.

**Lemma 1.** Let $p$ and $q$ be two probability density functions defined on a space $\mathcal{S}$. If there exists a set $\mathcal{A} \subseteq \mathcal{S}$ of strictly positive measure such that $\forall \theta \in \mathcal{A}$, $q(\theta) < p(\theta)$, then there exists a set $\mathcal{B} \subseteq \mathcal{S}$ of strictly positive measure such that $\forall \theta' \in \mathcal{B}$, $q(\theta') > p(\theta')$.

*Proof.* We know that $\int_{\mathcal{S}} p = \int_{\mathcal{S}} q = 1$ or equivalently, using $\mathcal{S} = \mathcal{A} \cup \mathcal{B} = \{q > p\} \cup \{q \leq p\}$,

$$\int_{\mathcal{A}} q(\theta)\mathrm{d}\theta + \int_{\mathcal{B}} q(\theta)\mathrm{d}\theta = \int_{\mathcal{A}} p(\theta)\mathrm{d}\theta + \int_{\mathcal{B}} p(\theta)\mathrm{d}\theta = 1 .$$

By grouping the integrals over $\mathcal{A}$ on one side and the ones over $\mathcal{B}$ on the other, we get:

$$\int_{\mathcal{A}} q(\theta)\mathrm{d}\theta - \int_{\mathcal{A}} p(\theta)\mathrm{d}\theta = \int_{\mathcal{B}} p(\theta)\mathrm{d}\theta - \int_{\mathcal{B}} q(\theta)\mathrm{d}\theta > 0 .$$

which is non-negative because $\mathcal{A}$ is assumed to be of strictly positive measure and $q - p > 0$ everywhere in $\mathcal{A}$.

The integral of $p - q$ over $\{p = q\}$ is zero, which implies that

$$\int_{q<p} \Big(p(\theta) - q(\theta)\Big) \mathrm{d}\theta = \int_{\mathcal{B}=q\leq p} \Big(p(\theta) - q(\theta)\Big) \mathrm{d}\theta > 0$$

meaning that the region $\{\theta \in \mathcal{S}, p(\theta) < q(\theta)\}$ is is of strictly positive measure, which concludes the proof.

## A.2 Proof of Theorem 2

**Theorem 2** (Local consistency and regression). *If $f_{x_o}$ is Bayes optimal and $N_v \to \infty$, then $\hat{t}_{\mathrm{MSE}}(f_{x_o}) = 0$ is a necessary and sufficient condition for the local consistency of $q$ at $x_o$.*

*Proof.* Let $d_{x_o}$ be an estimator of $\mathbb{P}(C = 1 \mid \Theta; x_o)$ defined on $\mathcal{S}_{x_o}$ such that $f_{x_o} = \mathbb{I}_{d_{x_o}>\frac{1}{2}}$. As this region contains all the data points $\Theta_n^q \sim q(\theta \mid x_o)$ $(C_n = 0)$ and $\Theta_n^p \sim p(\theta \mid x_o)$ $(C_n = 1)$, the mean squared error $\hat{t}_{\mathrm{MSE}}(f_{x_o})$ over the dataset $\mathcal{D} = \{(\Theta_n, C_n)\}_{n=1}^{2N}$ is well defined (9) and

$$\hat{t}_{\mathrm{MSE}}(f_{x_o}) \xrightarrow[N_v \to \infty]{} t_{\mathrm{MSE}}(f_{x_o}) = \frac{1}{2} \int \Big(d_{x_o}(\theta) - \frac{1}{2}\Big)^2 \Big(q(\theta \mid x_o) + p(\theta \mid x_o)\Big) \mathrm{d}\theta \ .$$

This integral is zero if, and only if $\big(d_{x_o}(\theta) - \frac{1}{2}\big)^2 = 0$ for every $\theta \in \mathcal{S}_{x_o}$ (all terms are non-negative and $q(\theta \mid x_o) + p(\theta \mid x_o) > 0)^9$, which is equivalent to $d_{x_o}(\theta) = \frac{1}{2}$ for every $\theta \in \mathcal{S}_{x_o}$. Assuming $f_{x_o} = f_{x_o}^\star$ is *optimal*, we have that $d_{x_o}(\theta) = d_{x_o}^*(\theta) = \mathrm{P}(C = 1 \mid \theta; x_o)$ and we conclude the proof using the result from equation (5) (and knowing that outside of $\mathcal{S}_{x_o}$, $p = q = 0$):

$$t_{\mathrm{MSE}}(f_{x_o}^\star) = 0 \quad \Longleftrightarrow \quad d_{x_o}^\star(\theta) = \mathrm{P}(C = 1 \mid \theta; x_o) = \frac{1}{2}, \forall \theta \in \mathcal{S}_{x_o} \quad \underset{(5)}{\Longleftrightarrow} \quad \mathcal{H}_0(x_o) \quad \text{holds} \ .$$

## A.3 Proof of Theorem 3

**Theorem 3** (Local consistency and single class evaluation). *If $f$ is Bayes optimal and $N_v \to \infty$, then $\hat{t}_{\mathrm{MSE}_0}(f, x_o) = 0$ is a necessary and sufficient condition for the local consistency of $q$ at $x_o$.*

*Proof.* Let $d$ be an estimator of $\mathbb{P}(C = 1 \mid \Theta; X)$ and $f = \mathbb{I}_{d>\frac{1}{2}}$ the associated classifier, both defined on $\mathcal{S} = \{w = (\theta, x) \in R^m \times \mathbb{R}^d, q(\theta, x) + p(\theta, x) > 0\}$ . Suppose that $f = f^\star$ is *optimal* and let $x_o \in \mathbb{R}^d$ be a *fixed* observation. As explained in section 3, we have that

$$d^\star(\theta, x_o) = \mathbb{P}(C = 1 \mid \theta; x_o) = d_{x_o}^\star(\theta) \quad \text{and} \quad f^\star(\theta, x_o) = f_{x_o}^\star(\theta), \quad \forall \theta \in \mathcal{S}_{x_o} \ .$$

Consider the support $\mathcal{S}_{q,x_o} = \{\theta \in \mathbb{R}^m, \quad q(\theta \mid x_o) > 0\} \subset \mathcal{S}_{x_o}$ containing all data points $\Theta_n^q \sim q(\theta \mid x_o)$ from our single-class validation set $\mathcal{D}_{v0} = \{(\Theta_n^q, 0)\}_{n=1}^{N_{v0}}$. Therefore our single-class test statistic $\hat{t}_{\mathrm{MSE}_0}$ is well defined (12) and

$$\hat{t}_{\mathrm{MSE}_0}(f^\star, x_o) \xrightarrow[N_{v0} \to \infty]{} t_{\mathrm{MSE}_0}(f^\star, x_o) = \int \Big(d_{x_o}^\star(\theta) - \frac{1}{2}\Big)^2 q(\theta \mid x_o) \mathrm{d}\theta$$

With the same arguments as in the proof of Theorem 2 in A.2, we get that

$$t_{\mathrm{MSE}_0}(f^\star, x_o) = 0 \quad \Longleftrightarrow \quad d_{x_o}^\star(\theta) = \mathbb{P}(C = 1 \mid \theta; x_o) = \frac{1}{2}, \quad \forall \theta \in \mathcal{S}_{q,x_o} \quad \Longleftrightarrow \quad \mathcal{H}_0(x_o) \quad \text{holds} \ ,$$

where the second equivalence is true because $\mathcal{S}_{q,x_o} = \mathcal{S}_{x_o}$ for $p(. \mid x_o) = q(. \mid x_o)$. Therefore, $t_{\mathrm{MSE}_0}(f^\star, x_o) = 0$ is a necessary and sufficient condition for $\mathcal{H}_0(x_o)$.

**N.B.** Single-class accuracy ($\mathrm{Acc}_0$) does not provide a sufficient condition for $\mathcal{H}_0(x_o)$.

---

$^9$Note that this integral can also be zero if $\mathcal{S}_{x_o}$ is of measure zero. But as mentioned at the beginning of this appendix, this has generally no practical implications.

*Proof.* Following the proof of Theorem 1 in A.1, we have that

$$\mathcal{H}_0(x_\mathrm{o}) \quad \text{holds} \iff \mathrm{Acc}(f^\star_{x_\mathrm{o}}) = \frac{1}{2} \iff \mathrm{P}(f^\star_{x_\mathrm{o}}(\Theta^q) = 0) + \mathbb{P}(f^\star_{x_\mathrm{o}}(\Theta^p) = 1) = 1 \;.$$

This means that $\mathrm{Acc}_0(f^\star, x_\mathrm{o}) = \mathrm{P}(f^\star_{x_\mathrm{o}}(\Theta^q) = 0) = \frac{1}{2}$ can only be a sufficient condition for $\mathcal{H}_0(x_\mathrm{o})$ if $\mathrm{P}(f^\star_{x_\mathrm{o}}(\Theta^q) = 0) = \mathbb{P}(f^\star_{x_\mathrm{o}}(\Theta^p) = 1)$, which is not generally true. In conclusion, evaluating $\mathrm{Acc}_0(f^\star, x_\mathrm{o})$ only provides a *necessary* condition for the local null hypothesis (see ($\Rightarrow$) in A.1).

# B Algorithms

## B.1 Generating PP-plots with $\ell$-C2ST

---

**Algorithm 5:** $\ell$-C2ST – local PP-plots for any $x_\text{o}$

---

**Input:** evaluation data set $\mathcal{D}_{v0}$; an observation $x_\text{o}$; estimate $d$ of the class probabilities; estimates $\{d_1, \ldots, d_{N_\mathcal{H}}\}$ under the null; grid $\mathcal{G}$ of PP-levels in (0,1); significance level $\alpha$

**Output:** empirical CDF-values $\{\hat{F}(l; x_\text{o})\}_{l \in \mathcal{G}}$ of predicted class-0 probabilities; $(1 - \alpha)$-confidence bands $\{L_l(x_\text{o}), U_l(x_\text{o})\}_{l \in \mathcal{G}}$

Predict class-0 probabilities $\{d_0(v, x_0) = 1 - d(v, x_\text{o})\}_{v \in \mathcal{D}_{v0}}$ /\* $d$ is an estimate of class 1 \hfill \*/

**for** $l$ *in* $\mathcal{G}$ **do**

    /\* Compute empirical CDFs at $l$ \hfill \*/

    $\hat{F}(l; x_\text{o}) \leftarrow \frac{1}{N_{v0}} \sum \mathbb{I}_{d_0(v, x_0) \leq l}$

    **for** $h = 1, \ldots, N_\mathcal{H}$ **do**

        $\hat{F}_h(l; x_\text{o}) \leftarrow \frac{1}{N_{v0}} \sum \mathbb{I}_{d_0(v, x_0) \leq l}$

    /\* Compute confidence bands at $l$ \hfill \*/

    $L_l(x_\text{o}), U_l(x_\text{o}) \leftarrow q_{\frac{\alpha}{2}}(\{\hat{F}_h(l; x_\text{o})\}_{h=1}^{N_\mathcal{H}}), q_{1 - \frac{\alpha}{2}}(\{\hat{F}_h(l; x_\text{o})\}_{h=1}^{N_\mathcal{H}})$ /\* quantiles \hfill \*/

**return** $\{\hat{F}(l; x_\text{o})\}_{l \in \mathcal{G}}, L_l(x_\text{o}), U_l(x_\text{o})\}_{l \in \mathcal{G}}$

---

## C   Run-times for each validation procedure

To complete the benchmark experiments from Section 4.1, we analyze the run-times of each validation method to compute the test statistic[10] for an NPE of the SLCP-task, obtained for different values of $N_{\text{train}}$. Results are displayed in Table 1 and computed on average over all given observations $x_{\text{o}}$ and for $N_{\text{cal}}$-values that ensure high test power. We here focus solely on the SLCP-task, as the higher dimensional observation space allows to illustrate the differences between local methods (trained on the joint data space, $(\theta, x) \in \mathbb{R}^{5+8}$) and the oracle (trained on the parameter space only, $\theta \in \mathbb{R}^5$).

As expected, the computation time increases with the sample size $N_{\text{cal}}$ (left vs. right part of Table 1). While the *oracle* C2ST has close to constant run-time for fixed $N_{\text{cal}}$, local methods become faster with increasing $N_{\text{train}}$. We observe that $\ell$-C2ST(-NF) has comparable run-times to the *oracle* C2ST: the amortization cost is negligible, in particular for difficult tasks involving "good" posterior estimators (high $N_{\text{train}}$). However, this is not the case for *local*-HPD, which is the most expensive method. Indeed, it involves (1) the costly computation of the HPD statistics on the joint and (2) the training of *several* (default is $n_c = 11$) classifiers, both of which increase with the sample size and dimension of the data space.

| $N_{\text{train}}$ | $N_{\text{cal}} = 5\,000$ | | | | $N_{\text{cal}} = 10\,000$ | | | |
|---|---|---|---|---|---|---|---|---|
| | $10^2$ | $10^3$ | $10^4$ | $10^5$ | $10^2$ | $10^3$ | $10^4$ | $10^5$ |
| *oracle* C2ST | **5.47** | **4.52** | 5.95 | 7.56 | **16.36** | **18.03** | 23.3 | 15.33 |
| $\ell$-C2ST | 5.92 | 5.09 | **1.78** | 1.81 | 27.62 | 34.06 | **17.9** | **3.65** |
| $\ell$-C2ST-NF | 6.98 | 6.84 | 6.38 | **1.72** | 43.99 | 25.01 | 18.56 | 7.62 |
| *local*-HPD | 282.19 | 282.85 | 279.38 | 282.5 | 956.91 | 938.21 | 682.92 | 530.45 |

Table 1: Run-time (in seconds) to compute the test statistic for the SLCP task (mean over observations). C2ST has close to constant run-time for fixed $N_{\text{cal}}$. Local methods become faster with increasing $N_{\text{train}}$ and $\ell$-C2ST(-NF) stays comparable to the *oracle* C2ST, even for $N_{\text{cal}} = 10\,000$. While the amortization cost of $\ell$-C2ST is not an issue, *local*-HPD is always at least 30 times slower.

---

[10]Note that in order to compute to compute p-values, we need to compute the test statistic $N_{\mathcal{H}}$ times under the null hypothesis. The number of classifiers we need to train depends on how many we need to compute the test statistic ($N_{\mathcal{H}}$ for $\ell$-C2ST vs. $N_{\mathcal{H}} \times n_c$ for *local*-HPD). In summary, if $\ell$-C2ST is more efficient in computing a single test statistic, it will also be more efficient to compute $N_{\mathcal{H}}$ test statistics.

# D Graphical diagnostics ($\ell$-C2ST-NF) for the JRNMM posterior estimator

The following figures present additional results for the interpretability of $\ell$-C2ST applied to the JRNMM neural posterior estimator. They complete Figure 4. Results are shown for all given observations associated to ground-truth gain values $g_\mathrm{o} = -20, -15, -10, -5, 0, 5, 10, 15, 20$.

In each Figure, the top right panel displays the empirical CDF of the classifier (blue) overlaid with the theoretical CDF of the null hypothesis (step function at $0.5$) and $95\%$ confidence region of estimated classifiers under the null displayed in gray. The pairplot displays histograms of samples from the posterior estimator $q_\phi$ within the prior region and dashed lines indicate values of $\theta_\mathrm{o}$ used to generate the conditioning observation $x_\mathrm{o}$. The predicted probabilities are mapped on the colors of the bins in the histogram. Blue-green (resp. orange-red) regions indicate low (resp. high) predicted probabilities of the classifier. Yellow regions correspond to chance level, thus $q_\phi \approx p$.

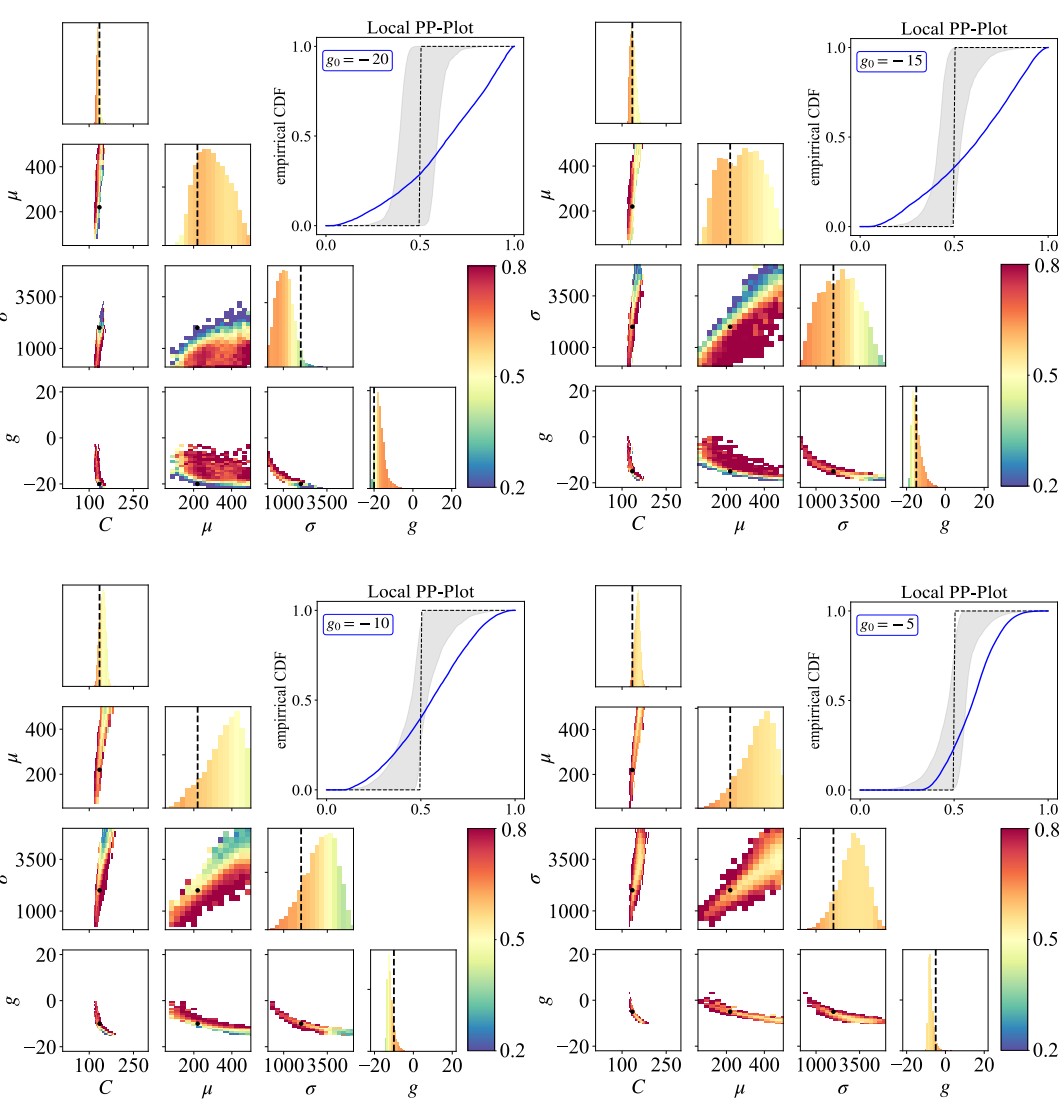

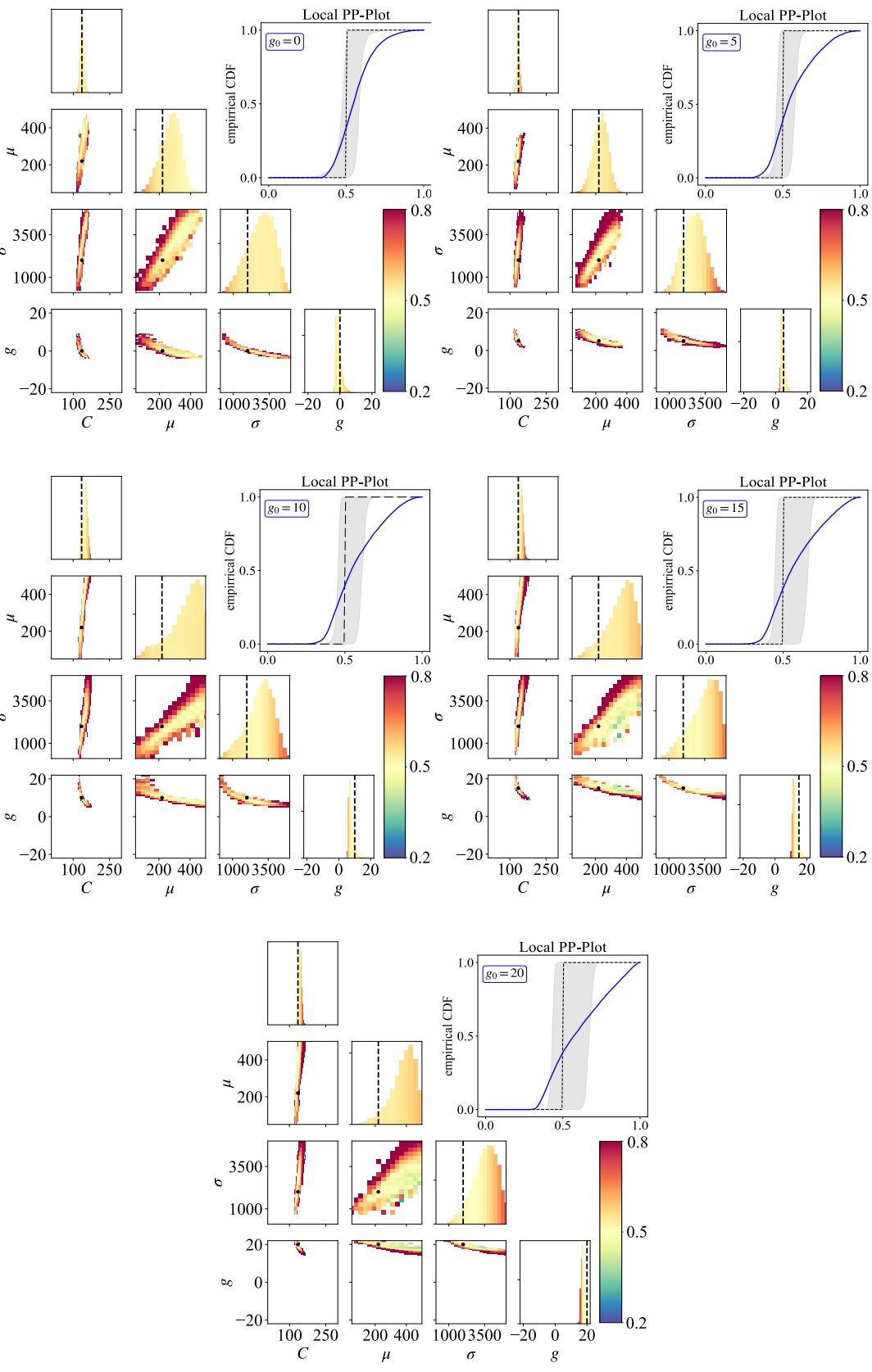

# E   On the cross-entropy loss for $\ell$-C2ST

This section aims at facilitating the understanding of $\ell$-C2ST by proving the result of equation (11).

As detailed in section 3, the first step in $\ell$-C2ST consists in training a classifier to distinguish between $N_{\text{cal}}$ data points $(\Theta_n, X_n)$ and $(\Theta_n^q, X_n)$ from the joint distributions $p(\theta, x)$ and $q(\theta, x)$ respectively. Here, the same conditioning observations $\{X_n\}_{n=1}^{N_{\text{cal}}}$ are used to construct the data samples for each class (see Algorithm 1). We show that this does not affect the objective function and convergence of the classifier.

The theoretical cross-entropy loss function to distinguish between data $(\Theta, X)$ from class $C = 1$ and class $C = 0$ is defined by

$$l_{\text{CE}}(d) := -\frac{1}{2}\mathbb{E}_{(\Theta, X)|C=1}\left[\log\left(d(\Theta, X)\right)\right] - \frac{1}{2}\mathbb{E}_{(\Theta, X)|C=0}\left[\log\left(1 - d(\Theta, X)\right)\right] \quad . \tag{20}$$

Note that the we have equal marginals $X \mid (C = 1) \sim p(x) = q(x) = X \mid (C = 0)$.[11] This allows us to take the expectation over $X$ and approximate (20) via Monte-Carlo for only one set of conditioning observations $\{X_n\}_{n=1}^{N_{\text{cal}}}$, and with data points $\Theta_n$ and $\Theta_n^q$ respectively associated to class $C = 1$ and $C = 0$ for a given $X_n$:

$$l_{\text{CE}}(d) = \mathbb{E}_X\left[-\frac{1}{2}\mathbb{E}_{\Theta|X,C=1}\left[\log\left(d(\Theta, X)\right)\right] - \frac{1}{2}\mathbb{E}_{\Theta|X,C=0}\left[\log\left(1 - d(\Theta, X)\right)\right]\right]$$
$$\approx -\frac{1}{2N_{\text{cal}}}\sum_{n=1}^{N_{\text{cal}}}\log\left(d(\Theta_n, X_n)\right) + \log\left(1 - d(\Theta_n^q, X_n)\right) \quad . \tag{21}$$

Therefore, we can train a classifier to minimize (20) using data from the joint distributions with same conditioning observations. The obtained estimate $d = \arg\min\{l_{\text{CE}}(d)\}$ of the class probabilities is defined for every $(\theta, x) \in \mathbb{R}^m \times \mathbb{R}^d$ by $d(\theta, x) \approx \frac{p(\theta, x)}{p(\theta, x) + q(\theta, x)}$. As $p(x) = q(x)$, we recover the class probabilities of the optimal Bayes classifier on the conditional data space for any given $x \in \mathbb{R}^d$ :

$$d^{\star}(\theta, x) = \frac{p(\theta \mid x)}{p(\theta \mid x) + q(\theta \mid x_{\text{o}})} = d_x^{\star}(\theta) \quad . \tag{22}$$

For an example, see works related to neural ratio estimation (NRE) [22, 8]: these algorithms implicitly use a classifier to distinguish between the joint and marginal distributions $p(\theta, x)$ and $p(\theta)p(x)$. Like in our case, both classes are modeled using the same observations $X_n$ obtained via the simulator.

---

[11]The joint distributions $p(\theta, x)$ and $q(\theta, x)$ are both modeled using samples $X \sim p(x \mid \Theta)$ obtained from the prior $\Theta \sim p(\theta)$, which implies that the marginals $p(x)$ and $q(x)$ are both defined by $\int p(x \mid \theta)p(\theta)\mathrm{d}\theta$.

# F  Additional Experiments on several benchmark examples

The results on the two benchmark examples in Figure 2 give first intuitions about the validity and behaviour of $\ell$-C2ST(-NF), our proposal, w.r.t. the *oracle* C2ST and the alternative *local*-HPD methods. In this section, Figure 5 extends Figure 2 with results on additional benchmark examples and more detailed results on the correlation between the test statistics of $\ell$-C2ST(-NF) and the oracle C2ST for different conditioning observations are shown in Figure 6 and Table 3. We refer the reader to Table 2 for a description of all benchmark examples (data dimensionality, posterior structure, challenges) and a summary of the main results comparing *local* methods.

While investigating the scalability of the algorithms to high dimensional data spaces, these additional experiments help to further analyze how well $\ell$-C2ST(-NF) captures the true C2ST, when it outperforms *local*-HPD, and better understand the benefit of the -NF version. These points are further detailed in the following subsections. The goal is to create a first guideline for when our proposal can and should be used, while raising awareness of its limitations (i.e. when it should *not* be used or at least be adpated for improved performance).

| | Dimension $(\theta, x)$ | Posterior structure | Challenge | Better *local* method |
|---|---|---|---|---|
| SLCP | low $(5, 8)$ | 4 symmetrical modes | complex posterior | $\ell$-**C2ST-NF** |
| Two Moons | low $(2, 2)$ | bi-modal, crescent shape | global and local structure | *local*-HPD / $\ell$-**C2ST-NF** |
| Gaussian Mixture | low $(2, 2)$ | 2D Gaussian | large vs. small s.t.d. in GMM | *all similar* |
| Gaussian Linear Uniform | medium $(10, 10)$ | multivariate Gaussian | dimensionality scaling | *local*-HPD / $\ell$-C2ST |
| Bernoulli GLM | medium $(10, 10)$ | unimodal, concave | dimensionality scaling | $\ell$-C2ST(**-NF**) |
| Bernoulli GLM Raw | high $(10, 100)$ | unimodal, concave | raw observations (no summary stats) | $\ell$-**C2ST-NF** |

Table 2: Description of benchmark examples and summary of main results for *local* methods.

## F.1  Scalability to high dimensions

First of all, note that in the specific case of SBI, the dimension of the parameter space is typically of order $10^0$ to $10^1$ and $m \approx 10^2$ is already often considered as high dimensional. The observation space, however, can be of higher dimension (e.g. $d \approx 10^3$ for time-series), but summary statistics are often used to reduce the dimension of the observations to the order of $d \approx 10^1$. In Section 4 we analyze the results obtained for rather low-dimensional benchmark examples (Two-Moons:$m = 2, d = 2$, SLCP: $m = 5, d = 8$). As an extension of Figure 2, Figure 5 includes additional benchmark examples with low and medium dimensionality: Gaussian Mixture ($m = 2, d = 2$) Gaussian Linear Uniform ($m = 10, d = 10$) and Bernoulli GLM ($m = 10, d = 10$). Furthermore, the Bernoulli GLM Raw task allows us to analyze how our method scales to high dimensional observation spaces only (without parameter-space / task variability): it considers raw observation data with $d = 100$, as opposed to sufficient summary statistics in the Bernoulli GLM task.

Column 3 in Figure 5 shows that $\ell$-C2ST requires more data to converge to the *oracle* C2ST (at maximum power TPR $= 1$) as the data dimensionality increases: $N_{\text{cal}} \approx 2000$ for Two Moons and SLCP, but $N_{\text{cal}} \approx 5000$ for the Bernouilli GLM task. Note that the Gaussian Mixture and

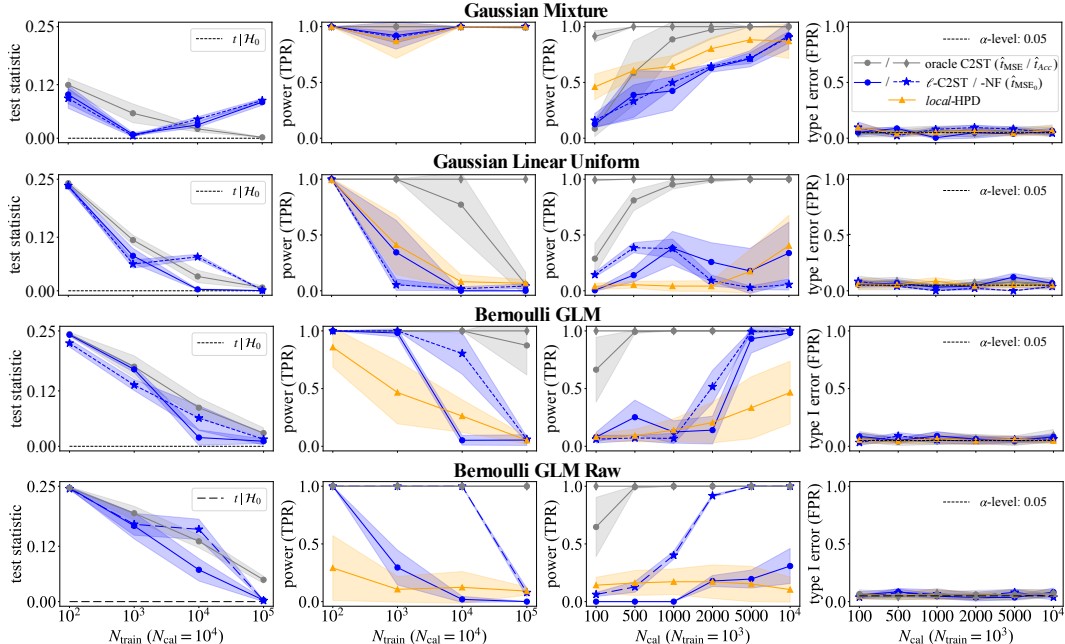

Figure 5: Results on additional `sbibm` benchmark examples using the same experimental setup as for Figure 2 in Section 4 (50 test runs, mean / std over 10 different reference observations).

`Gaussian Linear Uniform` tasks were not included in this analysis, as here, the difficulty of the classification task has more impact on statistical power than the data dimensionality [12].

Interestingly, we observe in the `Bernoulli GLM Raw` task, that $\ell$-C2ST-NF scales well to the high-dimensional observation space (faster convergence to maximum TPR compared to `Bernoulli GLM`), while the normal $\ell$-C2ST and local-HPD significantly lose in statistical power. It should also be noted that local-HPD performs significantly worse in medium dimensions (cf. `Bernoulli GLM` or even SLCP) than in low dimensions (`Gaussian Mixture` and `Two Moons`), though this could be because of the complex posterior structure.

### F.2   Accuracy of $\ell$-C2ST(-NF) w.r.t. the true C2ST

Figures 2 and 5 compare our method to the oracle C2ST, but only in terms of statistical power, as the local analysis is limited to the averaged results over 10 different reference observations $x_o$. We here provide a more detailed local analysis that examines how the $\ell$-C2ST(-NF) test statistics correlate with those from the *oracle* C2ST, by plotting them against each other. The results obtained for the above mentioned 10 reference observations are shown in Figure 6a. To allow for more robust conclusions, we also show results obtained for a total of 100 different reference observations (cf. Figure 6b), as well as quantitative results on the statistical significance of the correlation indices in Table 3.

Overall, the scattered points are not too far from the diagonal, which indicates that there is some correlation between the test statistics for $\ell$-C2ST(-NF) and those from the oracle C2ST. This correlation becomes weaker when $N_{\text{train}}$ becomes larger, since the test statistics in these cases tend to zero and can start to be confused with noise. This observation is consistent with the results in Table 3, showing the p-values of the Pearson test, a standard tests for the statistical significance of the correlation indices between the scores.

Another general trend is that the scattered points tend to be below the diagonal, indicating that $\ell$-C2ST(-NF) is less sensible to local inconsistencies than the *oracle* C2ST. This behaviour was expected, as $\ell$-C2ST is trained on the joint and thus less precise. Interestingly this doesn't apply to

---

[12]Local methods a TPR $< 1$ at $N_{\text{train}} = 1000$ (see Column 2), which means that the classification task is harder and requires more data: local methods never reach maximum TPR $= 1$, and even the *oracle* C2ST (with MSE test statistic) needs $N_{\text{cal}} = 2000$, at least four times more than for the other tasks.

the `Gaussian Mixture` task. This could be due to a big variability in the local consistency of $q$: while trained on the joint $\ell$-C2ST(-NF) could overfit on the "bad" observations, resulting in higher test statistics for observations where the true C2ST statistic would be small.

Finally, across all benchmark examples we observe results that are consistent with the ones from Figure 5: higher correlation means higher statistical power.

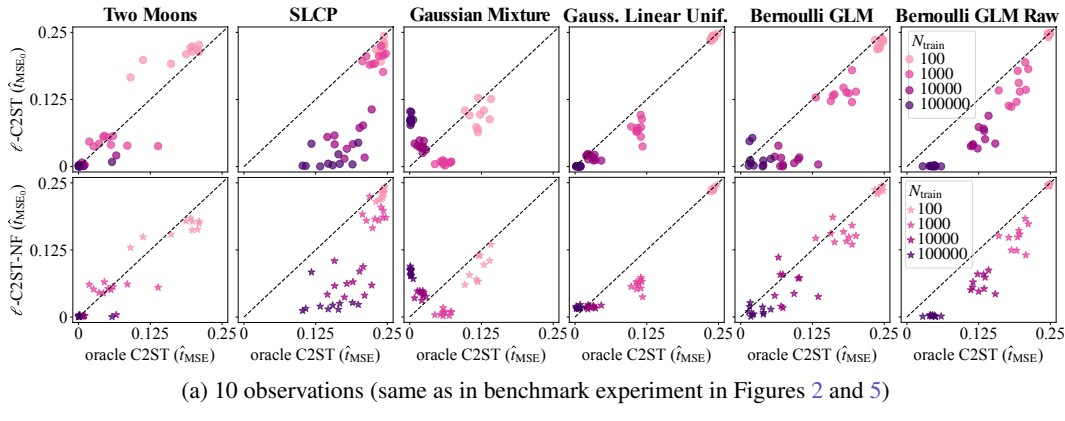

(a) 10 observations (same as in benchmark experiment in Figures 2 and 5)

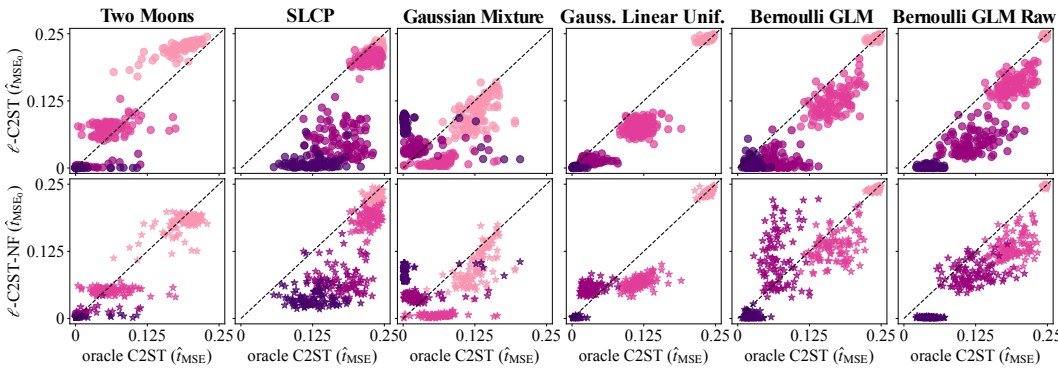

(b) 100 observations (generated via sbibm package)

Figure 6: Accuracy / correlation of $\ell$-C2ST(-NF) w.r.t. the *oracle* C2ST. We show scatter plots for all `sbibm` examples on (a) 10 and (b) 100 different reference observations. Each point corresponds to one observation and represents the MSE test statistic obtained for the oracle C2ST (x-axis) vs. our $\ell$-C2ST(-NF) method (y-axis). The diagonal represents the case where $\ell$-C2ST(-NF) = C2ST. The closer points are to the diagonal, the more accurate $\ell$-C2ST is w.r.t. C2ST.

|  | $N_{\text{train}}$ | | | |
|---|---|---|---|---|
|  | $10^2$ | $10^3$ | $10^4$ | $10^5$ |
| SLCP | $10^{-4}$ / 0.12 | $10^{-3}$ / 0.03 | 0.31 / 0.82 | 0.21 / 0.40 |
| Two Moons | $10^{-27}$ / $10^{-9}$ | $10^{-3}$ / 0.19 | $10^{-16}$ / $10^{-11}$ | 0.052 / $10^{-5}$ |
| Gaussian Mixture | $10^{-8}$ / $10^{-12}$ | $10^{-7}$ / 0.01 | 0.006 / 0.35 | $10^{-14}$ / 0.006 |
| Gauss. Linear Unif. | $10^{-13}$ / $10^{-12}$ | 0.07 / $10^{-9}$ | 0.42 / 0.002 | 0.68 / 0.87 |
| Bernoulli GLM | $10^{-8}$ / $10^{-5}$ | $10^{-10}$ / $10^{-4}$ | 0.67 / 0.18 | 0.39 / 0.31 |
| Bernoulli GLM Raw | 0.03 / 0.37 | $10^{-8}$ / $10^{-4}$ | $10^{-4}$ / 0.04 | 0.92 / 0.06 |

Table 3: P-values of the Pearson test of non-correlation between the $\ell$-C2ST( / -NF) and the *oracle* C2ST MSE test statistic. Obtained for 100 observations (as plotted in Figure 6b) using `scipy.stats.pearsonr`. Blue values indicate the cases for which the Pearson test rejects the null hypothesis of non-correlation with 95% confidence (i.e. there is significance evidence for correlation).

### F.3 Benefit of the -NF version

The results of all benchmark experiments indicate that the -NF version of $\ell$-C2ST works better when the (true) posterior distribution of the model is "more complicated" than a Gaussian distribution (see Table 2 at the beginning of Appendix F). This is for example the case for the `Two Moons` and SLCP tasks: the posterior distributions are globally multi-modal and locally structured. We observe in Column 3 of Figure 2 in the main paper that the -NF version requires less samples (i.e. lower $N_{\text{cal}}$) to reach maximum power/TPR. This is also the case for the additional `Bernoulli GLM` task (see Column 3 of Figure 5 in Appendix F.1). In contrast, for `Gaussian Mixture` and `Gaussian Linear Uniform`, where the posterior is Gaussian, the normal $\ell$-C2ST is as powerful or even better than its -NF counterpart.

Finally, we refer the reader to the analysis in Section F.1 to point out an interesting observation: for the `Bernoulli GLM`, the -NF version scales much better to high dimensional observation spaces than the normal $\ell$-C2ST. Note that this does not allow us to make any general conclusions, but it might be worth further investigating this result.

### F.4 Advantages of $\ell$-C2ST w.r.t. *local*-HPD

First of all, it is important to mention that having uniform HPD-values is not a sufficient condition for asserting the null hypothesis of consistency (see end of Section 3.3 in [48]). This is a clear disadvantage compared to our proposal, which provides a necessary and sufficient proxy for inspecting local posterior consistency.

Furthermore, the HPD methodology summarizes the whole information concerning $\theta$ into a single scalar, while in $\ell$-C2ST we handle the $\theta$-vector in its multivariate form. In medium-high $\theta$-dimensions ($m > 2$ as in `Bernoulli GLM`) or for complex posterior distributions (SLCP), such summarized information might discard too much information and not be enough to satisfactorily assess the consistency of the posterior estimator. Indeed, the tasks where *local*-HPD has similar statistical power to $\ell$-C2ST are either in low dimensions (`Two Moons`) and / or have a Gaussian posterior (`Gaussian Mixture` / `Gaussian Linear Uniform`). See Table 2 for a summary of those results w.r.t. data dimensionality and posterior structure. For a detailed analysis of the scalability to high dimensional data spaces see Appendix F.1.

Finally, *local*-HPD in its naive implementation is much less efficient than $\ell$-C2ST (see Appendix C). Note however that a new "amortized" version of *local*-HPD has recently been proposed [9] and could be interesting to look at.

