# OpenReview forum: "L-C2ST: Local Diagnostics for Posterior Approximations in Simulation-Based Inference"
_NeurIPS.cc/2023/Conference — NeurIPS 2023 poster_

### Official Review · Reviewer_Ppab · 2023-06-23

**Soundness:** 2 fair
**Presentation:** 4 excellent
**Contribution:** 3 good
**Rating:** 6
**Confidence:** 4

**Summary:**

This paper defines a new diagnostic to check posteriors learned from simulation-based inference.  The main issue in this setting is that one does not have access to likelihoods, and so one needs diagnostics that do not rely on being able to calculate likelihoods.  Here the authors present L-C2ST, which essentially tries to train a classifier to distinguish the _joint_ space over both parameters _and_ observations into whether a given point was more likely under the true joint distribution, $p(x, \theta)$ or under a variational approximation $q(\theta | x) p(x)$.  It turns out that a Bayes-optimal (probabilistic) classifier will assign equal weight to the two distributions for all $\theta$ for a given $x_0$ if and only if $q(\theta | x_0) = p(\theta | x_0)$ (up to some technical considerations).  As a result, the authors propose training such a classifier and then seeing how much its weights deviate from $0.5$ and using that as a measure of how well $q$ matches $p$.  Finally, the authors develop a hypothesis testing framework (to account for randomness in the trained classifier) to determine how significant a given deviation is.

**Strengths:**

* Trustworthy checks of how good a given variational posterior is for real settings where we do not have access to the true posterior are sorely needed, and this work is an important step in that direction.
* The paper is well-written, well-motivated, and clearly explained.
* The theoretical results complement and motivate the proposed algorithm well (but see below for some minor technical concerns).
* The proposed algorithm is extremely simple to describe, making it practical and elegant.

**Weaknesses:**

* My major concern is that I am not totally convinced that learning the proposed classifier is any easier than learning the variational posterior.  As such, I would be concerned that diagnostics showing a good posterior fit could be driven by a far-from-optimal classifier.  My intuition for why this should be a hard problem is as follows: for _any_ amortized posterior $q(\theta | x)$,  if we can learn a Bayes-optimal classifier $d^*$, then one could recover the true posterior for any $x$ via $p(\theta | x) = q(\theta | x) / (1-d^*(\theta, x))$.  As a result, learning $d^*$ for _any_ $q$ is equivalent to learning the true posterior.  This is somewhat borne out in the third column of figure 2, where one needs roughly the same number of samples to learn a good posterior as one needs in order to learn a good classifier.  I'm wondering if somehow the proposed approach is essentially equivalent to training two different amortized variational posteriors on independent data and then seeing how well they match.  Some further simulations showing that one problem is easier than the other would alleviate some of my concerns.  I should also say that I think that even if my concerns are valid, the diagnostic is still interesting and useful.
* I felt that the benchmarking could have been improved slightly.  In particular, some empirical relationship between the proposed measure of fit and existing measures would be nice.  For example, the authors could take the true posterior in a toy model where it is known, generate $q$ by perturbing that posterior in a systematic way and compare how increasingly large perturbations affect both the L-C2ST measure as well as things like KL, reverse KL, TV, and so on.  The results in Figures 1 and 2 hint at this, but there are a lot of moving parts (especially in Figure 2) making it difficult to determine whether to contribute power (or lack thereof) to failure of the SBI (i.e., $q$ is bad) or the classifier (i.e. $r$ is good).
* A minor technical point throughout the proofs of the theorems is that they rely on the null hypothesis $\mathcal{H}_0$ holding on a set of strictly positive measure.  That is, if $\mathcal{H}_0$ fails on a set of measure zero then the proofs would break down.  In particular, in that case $\mathcal{A}$ would have measure zero and the final integral after line 454 would be exactly zero.  I tend to feel that failures on sets of measure zero have no practical implications, and so this is a minor point. But, if one is going to state theorems and proofs, it would be good to be rigorous.
* Similarly, Lemma 1 should not be phrased in terms of "points" as those would have zero measure, and instead should say "If there exists a set $\mathcal{A} \subseteq \mathcal{S}$ with positive measure such that $p(\theta) > q(\theta)$ for all $\theta \in \mathcal{A}$, then there exists a set $\mathcal{B}$ with positive measure such that $p(\theta') < q(\theta')$ for all $\theta'\in \mathcal{B}$.
* Finally, there are similar  (and equally minor) issues appearing in the proofs of Theorems 2 and 3.

Typos:
* Line 103: "may lack of" --> "may lack"
* Line 240: "in average" --> "on average"
* Line 251: Missing period before "The local PP-plot"

**Questions:**

* Using the same $X_n$'s in Algorithm 1 for draws from $p(\cdot | X_n)$ and $q(\cdot | X_N)$ introduces dependencies in the training data for the classifier.  Can that somehow be leveraged when training the classifier (similar to how in linear regression, we would want to perform GLS instead of OLS to increase the power when data points have shared structure like this).  Along similar lines, if it is much cheaper to sample from $q(\theta | x)$ than it is to sample from $p(x, \theta)$, then could it help to somehow incorporate more samples from $q(\theta  | X_n)$ in the algorithm (or perhaps the whole distribution somehow)?  Obviously this would require accounting for the dependencies (perhaps via some kind of downweighting) in the training of the classifier.  This is obviously not necessary to implement for this paper, but I am curious to hear the authors thoughts.
* Algorithm 1 also seems computationally intensive due to the permutation step.  Would it be possible to do something like split-sample conformal inference to speed up the p-value calculations?  Again, obviously not necessary for the paper, I'm just curious.


**Limitations:**

I don't see any potential for direct societal harm from this work.  The authors may wish to include a limitations section that highlights failure modes of their method.

---

> ### Author Rebuttal · Authors · 2023-08-09
>
> We thank the reviewer for saying that our paper is well written and clearly explained. We also appreciate that he found our method "simple and elegant" and recognized that we aim at an important practical question for simulation-based inference. Here below we address some of the remarks and questions raised by the reviewer, which were very interesting and pointed towards exciting perspectives that we shall investigate in the near future.
>
> **Remark 1: My major concern is that I am not totally convinced that learning the proposed classifier is any easier than learning the variational posterior.**
>
> This is an interesting and true remark. I am not sure whether one tool could be considered as strictly easier than another. The results in Figure 2 of the initial submission indeed show that the sample-sizes are roughly the same for the `Two Moons` task. For the `SlCP` task however, the oracle C2ST statistic never reaches close-to-zero values, implying that much more samples (N_train) are needed for the posterior estimator to converge, than for $\ell$-C2ST to reach maximum TPR (N_cal). Results for additional benchmarks can be found in Figure 1 of the attached PDF: for the `Gaussian Linear Uniform` example classification appears harder than posterior estimation, for the `Bernoulli GLM` task however the posterior estimator appears to converge at $N_{\mathrm{train}}=10^5$ (see oracle C2ST curve in Column 1), while the maximum TPR is reached for $N_{\mathrm{cal}}=10^4$ (see $\ell$-C2ST-NF curve in Column 3).
>
> Being concerned about false conclusions due to a far-from-optimal classifier is totally justified. One should always make sure the classifier is “good enough” before using it as a diagnostic tool. If it’s not, using another diagnostic tool would be better. One way to do this would be to use cross-validation as a tool to select the classifier with highest accuracy for the observed data (the accuracy should never be exactly at 0.5) . Note, however, that the MSE test statistic for $\ell$-C2ST is defined by the predicted class probabilities and not the accuracy of the classifier. Therefore one should also check how well the classifier is calibrated.
>
> We have thought about this aspect of training a second posterior estimate instead of using a classifier to assess the consistency of the first posterior estimator. However, we decided that this would be redundant and preferred using two different tools to play against each other.  Furthermore, the choice of a binary classifier was not really motivated by the “easy to learn” aspect, but mainly the popularity of binary classification, the fact that it’s easy to understand and has a much richer and stable literature than deep generative models. We think that this way our method will be more convincing for people from many different fields, instead of just the sbi / bayesian inference community.
>
>
> **Remark 2: I felt that the benchmarking could have been improved slightly. In particular, some empirical relationship between the proposed measure of fit and existing measures would be nice (...).**
>
> We duly note this remark and thank the reviewer for pointing it out. Indeed, it would be nice to have some easier to understand and interpretable experiments, where less factors have an influence on the results. Unfortunately we did not have the time to design a new experiment.
>
>
> **Remark 3: A minor technical point throughout the proofs of the theorems is that they rely on the null hypothesis holding on a set of strictly positive measure (...)**
>
> We will correct the demonstrations in our camera ready version. We thank the reviewer for pointing this out!
>
> **Question 1: Using the same $X_n$'s in Algorithm 1 for draws from $p(.|X_n)$ and $q(.|X_n)$ introduces dependencies in the training data for the classifier. Can that somehow be leveraged when training the classifier (...). Along similar lines, if it is much cheaper to sample from $q(\theta|x)$ than it is to sample from $p(x, \theta)$, then could it help to somehow incorporate more samples from $q(\theta|X_n)$ in the algorithm (or perhaps the whole distribution somehow)? (....)**
>
> Using the same $X_n$’s for draws from $p$ and $q$ is a sensible and interesting topic. In Appendix A.7 in the initial submission we address this and show that in this case the theoretical cross-entropy loss is the same as if the $X_n$’s were drawn independently.
> However, we have thought about adding more samples from $q$ to improve our method as a follow-up to this work. When considering Normalizing Flows as posterior estimators, we would even have access to the likelihood. That could definitely help. My concern is that the classification problem becomes unbalanced and we might overfit on data from $q$ w.r.t to data from $p$. Interestingly the recent work Discriminative Calibration  from Yoa and Domke (2023) proposes an extension of C2ST for SBI that tries to leverage additional information of $q$, but without being *local* (main contribution of $\ell$-C2ST).
>
> **Question 2: Algorithm 1 also seems computationally intensive due to the permutation step. Would it be possible to do something like split-sample conformal inference to speed up the p-value calculations?**
>
> This is a very interesting question. We have taken a look at conformal inference to include a calibration step in the predict-prob from the classifiers. Indeed, using a  technique with finite-sample guarantees based on ordered statistics could probably allow us to avoid the rather time consuming permutations in the null hypothesis. However, it should be noted that conformal prediction is known to have difficulties in providing intervals with fixed conditional coverage, so for our local procedure this could be a problem. We will take a further look on this in the near future.
>
> **Limitations:** The authors may wish to include a limitations section that highlights failure modes of their method.
>
> We will try to include one in the final camera ready version of this article.

---

> > ### Comment · Reviewer_Ppab · 2023-08-10
> >
> > Thank you very much for the detailed response.  I found the paper interesting and enjoyable, but I stand by my initial score.

---

### Official Review · Reviewer_ukES · 2023-07-02

**Soundness:** 3 good
**Presentation:** 2 fair
**Contribution:** 3 good
**Rating:** 5
**Confidence:** 4

**Summary:**

The authors present a new diagnostic tool for simulation-based inference. The method learns the C2ST without knowledge of the true posterior distribution. If the density estimator is a normalizing flow, the authors propose to perform the classification in latent space.

**Strengths:**

**Originality**: The method tackles and important issue for simulation-based inference. The method is new and potentially very useful. The extension for normalizing flows is interesting and creative. If L-C2ST works well, I could see this be adopted by the community.

**Quality**: The method is rigorously derived.

**Clarity**: Everything is explained well and with sufficient detail. Figures are clear.

**Weaknesses:**

**Quality**:
My main concern with this paper is that it does not sufficiently demonstrate how well the method actually works. In particular, I believe that the following crucial questions remain unanswered:

- Does the method indeed capture the true C2ST for different `x`?

For example, the (decent) TPRs shown in figure 2 could also be obtained if L-C2ST converges onto the average C2ST across x_o. The central claim of the authors is that L2CT is local, but this claim needs substantially more empirical evidence.

To address this concerns, I would (for example) suggest to add a scatter plot of L-C2ST vs true C2ST on a task where the ground truth is available by MCMC or analytically

- How well does the method scale with parameter and data dimensionality

Since C2ST is trained on theta and x as input, I could imagine it to not scale well to high-D x or theta (or require **even more** simulations). All tasks in the paper are low-D though. I suggest that the authors add an additional benchmark task with higher data and parameter dimensionality.

- Is the method actually better than local-HPD

On one task, L-C2ST performs better than local-HPD and worse on another one. Which method should be preferred? The paper also claims that local-HPD is much less efficient than C2ST but this is never empirically shown. In particular, while a naive implementation of local-HPD might be slow, amortizing over the confidence alpha should be trivial and make local-HPD as fast as C2ST. Please correct me if I am wrong or, otherwise, clarify this in the paper.

To address this, I recommend adding an analysis of the runtimes of the algorithms and ideally have more than only two benchmark tasks.

I believe that the above concerns are critical to the usefulness of the method, but I would be willing to significantly increase my score if these things are addressed.


**Questions:**

- Do I understand correctly that, for large N_H, one has to train the classifier multiple times? If yes, couldn’t this make the method less efficient than local-HPD?

**Limitations:**

The authors state limitations of their method.

---

> ### Author Rebuttal · Authors · 2023-08-09
>
> We thank the reviewer for recognizing the potential and usefulness of the method that we propose, as well as appreciating its extension to the case of normalizing flows. We also thank the reviewer for the very deep and interesting questions related to our method. We have followed the reviewer's suggestions and have added results on more SBI benchmark examples, as well as new experiments for analyzing our results (cf. Figures 1 and 2 in the attached PDF). These examples are described and the results summarized in the global response to all reviewers.
>
> **Question 1: Does the method indeed capture the true C2ST for different x?**
>
> Figure 2 in the submitted paper compares our method to the oracle C2ST, but only in terms of statistical power, as we limited the *local* analysis to the averaged results over 10 different reference observations. Following the reviewer's suggestion, we've added scatter plots to examine how the values of the $\ell$-C2ST(-NF) test statistic correlate with those from the oracle C2ST. The results are presented in Figure 2.a of the attached PDF for the same 10 reference observations $x_o$ initially used. To allow for more robust conclusions, we've also included scatter plots using 100 reference observations (cf. Figure 2.b). Please note that we have also considered additional benchmark examples.
>
> Overall, the scattered points are not too far from the diagonal, which indicates that the test statistics for $\ell$-C2ST(-NF) correlate quite well with those from the oracle C2ST. Also, as $N_{train}$ increases, the scattered points become closer to zero and are more concentrated. This result comes with no surprise: when the posterior approximate is consistent, the test statistics should be close to zero. If the approximate is poorly estimated, the statistics should deviate from zero and the points can take on different values. We observe that the -NF version seems to be slightly closer to the oracle C2ST than the normal $\ell$-C2ST.
>
> In examples `Two Moons`, `Gaussian Linear Uniform`, and `Bernoulli GLM`, the scatterplots display points that are close to the diagonal. In tasks `SLCP` and `Bernoulli GLM Raw`, most scattered points are below the diagonal, especially for higher $N_{train}$ values. This means that $\ell$-C2ST has lower test static values, meaning that the null hypothesis will likely not be rejected: it is, therefore, less sensitive to local differences between $p$ and $q$ than the oracle C2ST, which is consistent with Figure 1 of the attached PDF. Intuitively, this can be explained by the fact that $\ell$-C2ST is trained on the joint pdf and is thus less precise.
>
> The results for the `Gaussian Mixture` task however deviate from the general trend, as in Figure 1 of the attached PDF. Maybe due to big variability in the *local* consistency of $q$. Unlike the true C2ST, $\ell$-C2ST is trained on the joint data space and could therefore overfit on the "bad" observations, resulting in higher test statistics for observations where the true C2ST statistic would be small.
>
> **Question 2: How well does the method scale with parameter and data dimensionality**
>
> Please see our response to Question 1 from Reviewer `Esye`.
>
> In our initial submission, the performance of C2ST on the SBI tasks that we considered was mainly impacted by the structure of the parameter space and the corresponding shape of the posterior distribution, since the examples were of rather low dimensionality. We have added more SBI tasks with varying sizes of observation/parameter space to give a clearer picture of the performance of our method.
>
> It should be noted that local-HPD performs significantly worse in medium dimensions (cf. `Bernoulli GLM` or even `SLCP`) than in low dimensions (`Gaussian Mixture` and `Two Moons`), though this could be because of the complex posterior structure. Also, $\ell$-C2ST-NF scales well to the high-dimensional observation space of `Bernoulli GLM`, while *local*-HPD significantly loses statistical power.
>
>
> **Question 3: Is the method actually better than local-HPD?**
>
> First of all, it is important to mention that having uniform HPD-values is not a sufficient condition for asserting the null hypothesis of consistency. This is mentioned by Zhao et al. (2021) (see end of section 3.3) and is a clear disadvantage compared to our proposal, which provides a necessary and sufficient proxy for inspecting local posterior consistency.
>
> Furthermore, the HPD methodology summarizes the whole information concerning $\theta$ into a single scalar, while in $\ell$-C2ST we handle the $\theta$-vector in its multivariate form. In high $\theta$-dimensions (`Bernoulli GLM`) or for complex posterior distributions (`SLCP`), such summarized information might discard too much information and not be enough to satisfactorily assess the consistency of the posterior estimator. Indeed, the only task where local-HPD outperforms $\ell$-C2ST is `Gaussian Mixture` (low dimension and gaussian posterior).
>
> Finally, as mentioned by the reviewer, local-HPD in its naive implementation is much less efficient than $\ell$-C2ST (see Appendix A.5 in submitted paper). A new version of *local*-HPD with amortized $\alpha$ has recently been proposed by DEY et al. (2023). However, we were not able to verify how well it works in time for the submission deadline.
>
> **Question 4: For large N_H, one has to train the classifier multiple times? (...) less efficient than local-HPD?**
>
> N_H is the number of times we compute the test statistic under the null hypothesis in order to compute p-values. The number of classifiers we need to train depends on how many we need to compute the test statistic ($N_H$ for $\ell$-C2ST vs. $N_H \times n_{\alpha}$   for local-HPD). In summary, if $\ell$-C2ST is more efficient in computing a single test statistic, it will also be more efficient to compute $N_H$  test statistics.

---

> > ### Comment · Reviewer_ukES · 2023-08-10
> > **Concerns on accuracy of l-C2ST**
> >
> > Thanks a lot for the detailed response and for the additional experiments, they helped in clarifying Q2, Q3 (although I still think that the gains over HPD are rather small), and Q4.
> >
> > Q1: However, I am worried about the results shown in the new pdf. In Figure 2, for any given training budget N_train, the oracle C2ST seems to correlate very weakly (if at all) with l-C2ST. How, then, should l-C2ST indeed be a **local** measure for calibration?

---

> > > ### Author Response · Authors · 2023-08-13
> > >
> > > Thank you for your response.
> > >
> > > **General remark on the performance of $\ell$-C2ST w.r.t. *local*-HPD:**
> > >
> > > Our goal with $\ell$-C2ST is to present a new, **alternative** method to *local*-HPD, which to our knowledge is the only other existing local diagnostic (please see **Remark 1** in the response to Reviewer `Ppab`, for further motivations regarding our choice to base our work on **C2ST**). Our research and experiments show that $\ell$-C2ST is theoretically valid and works on several datasets, sometimes even outperforming *local*-HPD. It is true that our method does not work as well on all examples, but a big advantage of $\ell$-C2ST is that one can directly use literature and advancements from the *classification* field in order to adapt and enhance it for any given dataset/task (e.g. see response to **Question 1** for Reviewer `Ppab`). This makes $\ell$-C2ST a competitive alternative with great potential.
> > >
> > > **Regarding the accuracy of $\ell$-C2ST and it actually being a **local** method:**
> > >
> > > - The method is based on solid mathematical reasoning and it is **local by definition**. Moreover, it has **if and only if** guarantees for consistency, a remarkable property for this kind of statistical test, which *local*-HPD does not have.
> > > - The suggested experiment with the scatterplots showing the "accuracy" of $\ell$-C2ST are really interesting! Our results in the attached PDF show that there is some correlation between the values of test statistics for $\ell$-C2ST and oracle-C2ST. This correlation becomes weaker when $N_{\mathrm{train}}$ becomes larger, since the test statistics in these cases tend to zero and can start to be confused with noise. We have carried out standard tests for the statistical significance of the correlation indices between the scores. The results are shown at the end of this comment.
> > > - The fact that $\ell$-C2ST does not track the exact values of the oracle C2ST on all examples and for every observation does not mean necessarily that it is not a good method for **detecting local posterior inconsistencies**. Our results on Figure 3 and Figure 4 (those on the `JRNMM` data) clearly show how the statistical test varies for each choice of observation $x_{\mathrm{o}}$. Also, it should be noted that while we want to be as close as possible to the oracle C2ST, the more adapted performance metric, measuring the capacity of **detecting local inconsitencies**, is the statistical error of the test (i.e. power and type 1 error): Figure 2 (resp. 1) in the main paper (resp. attached PDF) show that in more than one example, $\ell$-C2ST reaches maximum power  and outperforms *local*-HPD.
> > >
> > > **P-values of the Pearson test of non-correlation between the oracle C2ST and the $\ell$-C2ST( / -NF) MSE test statistic**.
> > > Obtained for 100 observations (as plotted in Figure 2.b.) using `scipy.stats.pearsonr`. Bold values indicate the cases for which the Pearson test rejects the null hypothesis of non-correlation with 95% confidence:
> > >
> > > `Two Moons`:
> > > - $N_{\mathrm{train}}=100$: **10e-27** / **10e-9**
> > > - $N_{\mathrm{train}}=1000$: **0.0006** / 0.19
> > > - $N_{\mathrm{train}}=10000$: **10e-16** / **10e-11**
> > > - $N_{\mathrm{train}}=100000$: 0.052 / **10e-5**
> > >
> > > `SLCP`:
> > > - $N_{\mathrm{train}}=100$: **0.0001** / 0.12
> > > - $N_{\mathrm{train}}=1000$: **0.0009** / **0.03**
> > > - $N_{\mathrm{train}}=10000$: 0.31 / 0.82
> > > - $N_{\mathrm{train}}=100000$: 0.21 / 0.40
> > >
> > >
> > > `Gaussian Mixture`:
> > > - $N_{\mathrm{train}}=100$: **10e-8** / **10e-12**
> > > - $N_{\mathrm{train}}=1000$: **10e-7** / **0.01**
> > > - $N_{\mathrm{train}}=10000$: **0.006** / 0.35
> > > - $N_{\mathrm{train}}=100000$: **10e-14** / **0.006**
> > >
> > > `Gaussian Linear Uniform`:
> > > - $N_{\mathrm{train}}=100$: **10e-13** / **10e-12**
> > > - $N_{\mathrm{train}}=1000$: 0.07 / **10e-9**
> > > - $N_{\mathrm{train}}=10000$: 0.42 / **0.002**
> > > - $N_{\mathrm{train}}=100000$: 0.68 / 0.87
> > >
> > > `Bernoulli GLM`:
> > > - $N_{\mathrm{train}}=100$: **10e-8** / **10e-5**
> > > - $N_{\mathrm{train}}=1000$: **10e-10** / **0.0002**
> > > - $N_{\mathrm{train}}=10000$: 0.67 / 0.18
> > > - $N_{\mathrm{train}}=100000$: 0.39 / 0.31
> > >
> > > `Bernoulli GLM Raw`:
> > > - $N_{\mathrm{train}}=100$: **0.03** / 0.37
> > > - $N_{\mathrm{train}}=1000$: **10e-8** / **0.0004**
> > > - $N_{\mathrm{train}}=100$: **0.0001** / **0.04**
> > > - $N_{\mathrm{train}}=100$: 0.92 / 0.06

---

> > > > ### Comment · Reviewer_ukES · 2023-08-14
> > > > **Quick response**
> > > >
> > > > (Longer response later, but) would you mind also sharing the correlation coefficient itself (not only the p-value and confidence interval). Thanks!

---

> > > > > ### Author Response · Authors · 2023-08-14
> > > > >
> > > > > Of course! Below you will find the **Pearson correlation coefficients** associated to the above p-values. This statistic varies between -1 and 1, 0 indicating zero-correlation between the oracle C2ST and $\ell$-C2ST( / -NF). In bold the cases for which the associated p-value is $<0.05$ (see above results).
> > > > >
> > > > > `Two Moons`:
> > > > > - $N_{\mathrm{train}}=100$: **0.84** / **0.57**
> > > > > - $N_{\mathrm{train}}=1000$: **0.34** / -0.13
> > > > > - $N_{\mathrm{train}}=10000$: **0.71** / **0.61**
> > > > > - $N_{\mathrm{train}}=100000$: 0.20 / **0.40**
> > > > >
> > > > > `SLCP`:
> > > > > - $N_{\mathrm{train}}=100$: **0.39** / 0.16
> > > > > - $N_{\mathrm{train}}=1000$: **0.34** / **0.23**
> > > > > - $N_{\mathrm{train}}=10000$: -0.11 / 0.02
> > > > > - $N_{\mathrm{train}}=100000$: 0.13 / -0.09
> > > > >
> > > > >
> > > > > `Gaussian Mixture`:
> > > > > - $N_{\mathrm{train}}=100$: **0.52** / **0.63**
> > > > > - $N_{\mathrm{train}}=1000$: **0.49** / **0.25**
> > > > > - $N_{\mathrm{train}}=10000$: **0.27** / 0.09
> > > > > - $N_{\mathrm{train}}=100000$: **-0.68** / **0.27**
> > > > >
> > > > > `Gaussian Linear Uniform`:
> > > > > - $N_{\mathrm{train}}=100$: **0.64** / **0.63**
> > > > > - $N_{\mathrm{train}}=1000$: 0.18 / **0.55**
> > > > > - $N_{\mathrm{train}}=10000$: 0.08 / **0.31**
> > > > > - $N_{\mathrm{train}}=100000$: -0.04 / 0.02
> > > > >
> > > > > `Bernoulli GLM`:
> > > > > - $N_{\mathrm{train}}=100$: **0.54** / **0.42**
> > > > > - $N_{\mathrm{train}}=1000$: **0.58** / **0.37**
> > > > > - $N_{\mathrm{train}}=10000$: -0.04 / 0.14
> > > > > - $N_{\mathrm{train}}=100000$: -0.09 / -0.10
> > > > >
> > > > > `Bernoulli GLM Raw`:
> > > > > - $N_{\mathrm{train}}=100$: **0.22** / 0.09
> > > > > - $N_{\mathrm{train}}=1000$: **0.53** / **0.35**
> > > > > - $N_{\mathrm{train}}=100$: **0.37** / **0.20**
> > > > > - $N_{\mathrm{train}}=100$: -0.01 / -0.19

---

> > > > > > ### Comment · Reviewer_ukES · 2023-08-14
> > > > > > **Response**
> > > > > >
> > > > > > Thanks for the additional explanation and experiments!
> > > > > >
> > > > > > Another quick question upfront: how large is $N_{cal}$ in the experiments above? I.e., how many simulations is the classifier trained on?
> > > > > >
> > > > > > Apart from this, I mostly agree with what the authors wrote, but I do disagree with this sentence:
> > > > > > ```[oracle C2ST not matching l-C2ST] does not mean necessarily that it is not a good method for detecting local posterior inconsistencies.```
> > > > > >
> > > > > > Of course, it does not have to track the oracle C2ST perfectly to be useful. I also agree that, even with a correlation coefficient of zero, the method might estimate the **average** C2ST and still have decent statistical power compared to HPD. However, I do think that the claim of the method being local is extremely misleading in scenarios where the correlation between the oracle C2ST and l-C2ST is almost 0. Some of the reported correlations are very low and warrant an extended paragraph on limitations of the method which highlights regimes in which the classifier does not report **meaningful** local C2ST estimates and essentially produces noisy samples of the average C2ST (or shows very weak correlation with the true oracle C2ST). For example, I believe that it might be useful to provide ballpark estimates of how large $N_{cal}$ must be in comparison to $N_{train}$ in order to reach a certain correlation coefficient.
> > > > > >
> > > > > > Nonetheless, I have increased my score to 5 because I believe that the method can be useful. I hope that, if the paper gets accepted or resubmitted, the authors will add the analyses from this review process to the paper and will add a discussion of the limitations on how faithful the estimated l-C2ST is. Nonetheless, congrats on nice (and hopefully useful) paper!

---

> > > > > > > ### Author Response · Authors · 2023-08-14
> > > > > > >
> > > > > > > Hello!
> > > > > > >
> > > > > > > Quick response: $N_{\mathrm{cal}} = 10^4$ in the experiment above and in all experiments where $N_{\mathrm{train}}$ is varying (from the main paper and the PDF attached to this rebuttal).
> > > > > > >
> > > > > > > Thank you for the constructive questions and remarks, they are much appreciated. We are glad that we could answer most of your questions and will discuss your last comments.
> > > > > > >
> > > > > > > We will make sure to include the additional experiments of this rebuttal, as well as a paragraph on the limitations / failure modes of $\ell$-C2ST!

---

### Official Review · Reviewer_Esye · 2023-07-04

**Soundness:** 4 excellent
**Presentation:** 4 excellent
**Contribution:** 3 good
**Rating:** 7
**Confidence:** 2

**Summary:**

An algorithm for evaluating amortized posterior estimators $q(\theta|x)$ in simulation-based inference (SBI) is proposed. In the setting considered, we have the ability to sample $p(\theta)$ and to sample $p(x|\theta)$ but not to evaluate its density. The algorithm modifies the common classifier two-sample test by learning to discriminate between samples $(\theta,x)$ sampled ancestrally (though simulation) from $p(\theta,x)$ and those where $\theta$ is then resampled from $q(\theta|x)$. An extension to normalizing flows is also considered, where the discriminator works in the latent space rather than in the theta space. There are strong results on several SBI benchmarks and real-world problems in terms of statistical power and runtime.

**Strengths:**

Please note that this is a relative outsider's point of view, as I do not directly work on SBI.

I find the paper well-written and was able to understand the main points and the math. In particular, the first few pages are a good introduction to the problem area that will be accessible to any reader who is familiar with Bayesian inference and hypothesis testing.

The results and their significance are explained well and give intuitions about why the proposed algorithm works well on the chosen problems.

The provided code is a helpful addition.

**Weaknesses:**

I do not see any major weaknesses, but have a few questions.
- Could you please comment on the scalability of the algorithm? What difficulties do you foresee when scaling to high-dimensional theta or problems when the prior is wide? (Same question about high-dimensional observations. Is more risk of overfitting to the class split of $x$?)
- It would be interesting to understand better the benefit of the -NF version of the algorithm beyond the reasons described in L180, which seem to explain only computation cost improvements. Is it expected that the classification boundary is smoother in the latent space than in the theta space? Maybe this could be illustrated, e.g., on the two moons example.

**Questions:**

Please see above.

**Limitations:**

yes

---

> ### Author Rebuttal · Authors · 2023-08-09
>
> We thank the reviewer for finding our paper "clear and well-written", as well for having checked our code. We now answer the two main questions raised by the reviewer. Please note that we have added results on more SBI benchmark examples that you will find in Figures 1 and 2 of the PDF attached to the rebuttal. These examples are described and the results summarized in the global response to all reviewers.
>
>
> **Question 1: Scalability of the algorithm to high-dimensional theta or x**
>
> An obvious answer would be to say that in higher dimensions more samples are needed: larger $N_{\mathrm{cal}}$ for $\ell$-C2ST(-NF) to converge to the oracle C2ST. In our setting, this would mean better convergence to the *optimal Bayes classifier*, which is known to be the same for $\ell$-C2ST and oracle C2ST. Note, however, that the MSE test statistic for $\ell$-C2ST is defined by the predicted class probabilities and not the accuracy of the classifier. Therefore, adding a regularizer or maybe a calibration step (e.g. using methods from the conformal inference literature) might be necessary when working with data defined in high dimensions, so as to prevent overfitting and overconfident predictions.
>
> In the specific case of SBI, the dimension of the parameter space ($m$) is typically of order $10^0$ to $10^1$ and $m \approx 10^2$ is already often considered as high dimensional. The observation space, however, can be high-dimensional (e.g. time-series), but summary statistics are often used to reduce the dimension of the observations to the order of $d \approx 10^1$.
>
> In our initial submission, we show results for rather low-dimensional tasks ($m=2, d=2$) for Two-Moons and $m=5, d=8$ for `SLCP`). They illustrate how our method behaves in terms of the difficulty of the inference task (e.g. complex posteriors as in `SLCP`). To demonstrate how the results change with respect to the dimensionality of the data, we've added more tasks of low and medium dimensionality: `Gaussian Mixture` ($m=2, d=2$) `Gaussian Linear Uniform` ($m=10, d=10$) and `Bernoulli GLM` ($m=10, d=10$). To analyze how our method scales to **high dimensional observation spaces** only (without parameter-space / task variability), we’ve also added the `Bernoulli GLM Raw` task.  It considers *raw* observation data ($d=100$), as opposed to *sufficient summary statistics* in `Bernoulli GLM` ($d=10$). We refer the reader to the global response for all reviewers for a description of all benchmark examples (in terms of dimensionality, posterior structure, challenges) and a summary of all experimental results.
>
> In the attached PDF , Figure 1 shows results obtained for the new benchmarks, therefore extending Figure 2 of the initial submission. We can see in Column 3 of both Figures that  for $\ell$-C2ST to converge to the oracle C2ST (at maximum power $\mathrm{TPR} = 1$), less samples are needed in the low-dimensional `Two Moons` and `SLCP` tasks than for the medium-dimensional `Bernoulli GLM` task: $N_{\mathrm{cal}} \approx 2000$ vs. $N_{\mathrm{cal}} \approx 5000$. This confirms our intuition stated above.
>
> Note that we did not include the `Gaussian Mixture` and `Gaussian Linear Uniform` tasks in this analysis, as they are not comparable to the other tasks: the TPR of $\ell$-C2ST at $N_{\mathrm{train}}=1000$ is smaller than 1 (see Column 2). The classification task is thus harder and more samples are required to converge to reach maximum TPR: the oracle C2ST now requires $N_{\mathrm{cal}}=2000$ and *local* methods never reach maximum TPR (see Column 3). Here, the difficulty of the classification task has more impact on statistical power than the dimensionality of the task (e.g. the convergence to maximum TPR is slower in `Gaussian Mixture` than in `Bernoulli GLM` of higher dimension).
>
> Interestingly, we observe in the  `Bernoulli GLM Raw` task, that $\ell$-C2ST-NF scales well to the high-dimensional observation space (faster convergence to maximum TPR compared to the `Bernoulli GLM` task), while the normal $\ell$-C2ST and *local*-HPD significantly lose in statistical power.
>
>
> **Question 2: Benefit of the -NF version**
>
> The numerical illustrations in our manuscript indicate that the -NF version of our statistical test works better when the (true) posterior distribution of the model is "more complicated" than a gaussian distribution. This is the case for the `Two Moons` and `SLCP` tasks: the posterior distributions are globally multi-modal and locally structured (cf. task descriptions in the global response to all reviewers). We observe in Column 3 of Figure 2 in the main paper that the -NF version requires **less samples** (i.e. lower $N_{\mathrm{cal}}$) to reach maximum power/TPR. This is also the case for the additional `Bernoulli GLM` task (see Column 3 of Figure 1 in the attached PDF). In contrast, the additional `Gaussian Mixture`and `Gaussian Linear Uniform` tasks, where the posterior is a gaussian distribution, the normal $\ell$-C2ST is as powerful or even better than its -NF counterpart (see Column 3 of Figure 1 in the attached PDF).
>
> We have also observed that $\ell$-C2ST-NF yields test statistics that correlate more closely with the oracle C2ST in situations where the true posterior distribution can be sampled with MCMC. The experiments and illustrations related to this finding can be found in Figure 2 of the attached PDF and are described in our response to Reviewer `ukES02`.
>
> Finally, we refer the reader to the previous question to point out an interesting observation: for the `Bernoulli GLM`, the -NF version scales much better to high dimensional observation spaces than the normal $\ell$-C2ST. Note that this does not allow us to make any general conclusions, but it might be worth further investigating this result.

---

### Author Rebuttal · Authors · 2023-08-09

We thank the reviewers for their thorough reading of our manuscript and their very interesting questions. We have had very positive remarks concerning the clarity of our text and the potential of our method to the SBI community. We are very grateful for all this. Some interesting questions that were raised concerned how our method behaves when the dimensionality of the data increases, whether the test statistics of $\ell$-C2ST really capture the true test statistics (i.e. those from the oracle C2ST) for different values of conditioning observation $x$, and whether our method is indeed superior to local HPD. We address all these questions (and some more) below.

We have followed suggestions and have added results on further examples from the SBI benchmark. In the attached PDF, Figure 1 extends Figure 2 of the submitted version, with the intent of better validating our method on varying dimensions for observations and parameters. We have also included results for a new experiment in Figure 2 of the PDF, showing the correlation between the test statistics of $\ell$-C2ST(-NF) and those from the oracle C2ST for different conditioning observations.


**Description of the used SBI benchmark examples from `sbibm`  and summary of main results:**

`Two Moons`:

- **Dimensions ($\theta, x$):** (2,2)
- **Posterior structure:** bi-modal, crescent shape
- **Challenge:** globally and locally structured

- **$\ell$-C2ST(-NF) vs. local-HPD**  (power, Figure 1): faster convergence to max power, but worse for high $N_{train}$
- **$\ell$-C2ST-NF vs. L-C2ST** (power, Figure 1): better (less samples for higher power)
- **$\ell$-C2ST(-NF) vs. C2ST** (relative position to diagonal, Figure 2): good

`SLCP`:

- **Dimensions ($\theta, x$):** (5,8)
- **Posterior structure:** 4 symmetrical modes
- **Challenge:** posterior designed to be complex

- **$\ell$-C2ST(-NF) vs. local-HPD**  (power, Figure 1): better
- **$\ell$-C2ST-NF vs. L-C2ST** (power, Figure 1): better
- **$\ell$-C2ST(-NF) vs. C2ST** (relative position to diagonal, Figure 2): lower

`Gaussian Mixture`:

- **Dimensions ($\theta, x$):** (2,2)
- **Posterior structure:** 2D gaussian
- **Challenge:** one of the gaussians in the mixture has much broader covariance than the other.

- **$\ell$-C2ST(-NF) vs. local-HPD**  (power, Figure 1): slightly worse
- **$\ell$-C2ST-NF vs. L-C2ST** (power, Figure 1): same
- **$\ell$-C2ST(-NF) vs. C2ST** (relative position to diagonal, Figure 2): lower, then higher

`Gaussian Linear Uniform`:

- **Dimensions ($\theta, x$):** (10,10)
- **Posterior structure:** multivariate Gaussian
- **Challenge:** scaling to higher dimensions

- **$\ell$-C2ST(-NF) vs. local-HPD**  (power, Figure 1):  similar
- **$\ell$-C2ST-NF vs. L-C2ST** (power, Figure 1): worse
- **$\ell$-C2ST(-NF) vs. C2ST** (relative position to diagonal, Figure 2): good

`Bernoulli GLM`:

- **Dimensions ($\theta, x$):** (10,10)
- **Posterior structure:** unimodal, concave
- **Challenge:** scaling to higher dimensions

- **$\ell$-C2ST(-NF) vs. local-HPD**  (power, Figure 1): better
- **$\ell$-C2ST-NF vs. L-C2ST** (power, Figure 1): better
- **$\ell$-C2ST(-NF) vs. C2ST** (relative position to diagonal, Figure 2): good / slightly lower

`Bernoulli GLM Raw`:

- **Dimensions ($\theta, x$):** (10,100)
- **Posterior structure:** unimodal, concave
- **Challenge:** Scaling to high dimensional observation spaces

- **$\ell$-C2ST(-NF) vs. local-HPD**  (power, Figure 1):  better
- **$\ell$-C2ST-NF vs. L-C2ST** (power, Figure 1): better
- **$\ell$-C2ST(-NF) vs. C2ST** (relative position to diagonal, Figure 2): slightly lower

---

### Decision · Program_Chairs · 2023-09-21

**Decision:**

Accept (poster)

**Comment:**

The paper introduces a diagnostic tool, L-C2ST, for evaluating the quality of posterior at any given observation in simulation-based inference (SBI). This is achieved by training a classifier to discern between samples from the true joint distribution and those from a variational approximation. The work delves into the Bayes-optimal classifier's significance in the context, and an hypothesis testing framework is developed to understand the deviation's significance.

**Strengths:**
- Novel diagnostic tool for simulation-based inference, tackling a significant gap in current SBI methods.
- Strong theoretical grounding that is explained well and complements the proposed method.
- The paper is well-written, making it accessible to readers not deeply familiar with SBI.
- Provides open-source code.

**Weaknesses:**
- Several reviewers raised concerns about the method's scalability with high-dimensional data or parameters, an aspect not sufficiently addressed in the paper.
- Concerns about the actual utility and practicality of training the classifier: Is it inherently easier than obtaining a good variational posterior?
- The paper doesn't provide enough evidence about the true performance of the method, especially in capturing the true C2ST for different x values and the comparison to existing methods like local-HPD.
- Benchmarking could be more extensive, possibly comparing the proposed diagnostic with other measures using a known true posterior.

The paper has significant potential and originality, though there remain concerns particularly in its empirical validation and scalability. Given its potential and the positive reviews, I recommend accept with a strong suggestion for revisions to address the highlighted weaknesses.